# Allosteric modulation of ghrelin receptor signaling by lipids

Marjorie Damian[1], Maxime Louet[1], Antoniel Augusto Severo Gomes[1,2], Céline M'Kadmi[1], Séverine Denoyelle[1], Sonia Cantel[1], Sophie Mary[1], Paulo M. Bisch[2], Jean-Alain Fehrentz[1], Laurent J. Catoire [3], Nicolas Floquet[1] & Jean-Louis Banères [1✉]

The membrane is an integral component of the G protein-coupled receptor signaling machinery. Here we demonstrate that lipids regulate the signaling efficacy and selectivity of the ghrelin receptor GHSR through specific interactions and bulk effects. We find that PIP2 shifts the conformational equilibrium of GHSR away from its inactive state, favoring basal and agonist-induced G protein activation. This occurs because of a preferential binding of PIP2 to specific intracellular sites in the receptor active state. Another lipid, GM3, also binds GHSR and favors G protein activation, but mostly in a ghrelin-dependent manner. Finally, we find that not only selective interactions but also the thickness of the bilayer reshapes the conformational repertoire of GHSR, with direct consequences on G protein selectivity. Taken together, this data illuminates the multifaceted role of the membrane components as allosteric modulators of how ghrelin signal could be propagated.

[1] IBMM, UMR 5247, CNRS, Université de Montpellier, ENSCM, Montpellier, France. [2] Laboratório de Física Biológica, Instituto de Biofísica Carlos Chagas Filho, Universidade Federal do Rio de Janeiro, Rio de Janeiro, RJ, Brazil. [3] Laboratoire de Biologie Physico-Chimique des Protéines Membranaires, UMR 7099, CNRS, Université de Paris, Institut de Biologie Physico-Chimique (FRC 550), Paris, France. ✉email: jean-louis.baneres@umontpellier.fr

G protein-coupled receptors (GPCRs) are integral membrane proteins that are prominent players in most intercellular communication events[1]. Upon agonist binding, GPCRs activate many downstream intracellular partners including different G protein subtypes and arrestins. The current model proposes that receptors explore complex conformational landscapes populated with multiple states of distinct functional properties, and that the relative distribution of these states is modulated by ligands, signaling proteins, and the environment, ultimately dictating the signaling output[2,3]. As such, GPCR conformational dynamics likely represents a driving force in the signaling process.

Lipids and membrane proteins have evolved together to provide a fully integrated system that fulfills exquisitely regulated functions. Hence, the cell membrane does not only provide the physiological environment necessary for the stability of the native fold of membrane proteins but it also modulates their function through an impact on their conformational dynamics. This makes the lipid and proteins intricated components of a single molecular machine[4]. Lipids affect membrane protein dynamics in different ways. One is related to the physicochemical properties of the bilayer, e.g., its curvature, lateral pressure, and thickness. In the case of GPCRs, the effect of membrane bulk properties has been essentially analyzed with rhodopsin where it has been shown to modulate the photocycle and dimerization propensity[5,6]. Besides, the composition of the membrane can affect GPCR functioning because of direct interactions with specific lipids such as cholesterol[7,8], phosphatidylinositol (4,5)-bisphosphate (PIP2)[9] or docosahexaenoic acid (DHA)[10]. However, although the development of analytical methods in membrane-like environments has advanced our current understanding of the functional relevance of lipids in GPCR signaling, experimental evidence for the relationship between membrane composition, receptor dynamics, and signaling behavior is scarce. Hence, most of our understanding still arises from computational studies[11].

Ghrelin is a 28-amino-acid gastrointestinal peptide hormone that exerts a wide range of biological effects through a single GPCR, the growth hormone secretagogue receptor (GHSR)[12]. These effects include the control of growth hormone secretion, food intake, glucose metabolism, and of response to reward and stress[12]. GHSR is a typical rhodopsin-like GPCR that signals through multiple pathways involving Gαq, Gαi2/o, Gα12/13, and arrestins[13,14]. This receptor is expressed in a variety of tissues and cell types, both at the central and peripheral levels[12]. As such, it likely experiences a variety of membrane environments that could impact on its pharmacological properties. Accordingly, several studies indicated that lipids affect the metabolic action of ghrelin[15,16] and demonstrated that the composition of the membrane has an impact on GHSR activation and desensitization[15].

We applied here a combination of experimental and computational methods to the purified GHSR assembled into nanodiscs to assess how the different features of the lipid bilayer could affect the structure and function of this receptor. To this end, we first analyzed the effect of the interaction of GHSR with two lipids, namely the phosphoinositide PIP2 and the glycosphingolipid GM3, that had been both proposed to interact with GPCRs[17]. PIP2 is a key element in the regulation of integral membrane proteins[18]. As a consequence, dysregulation in phosphatidylinositol synthesis, transport, or metabolism participates in human diseases involving membrane proteins misfunction[18]. Gangliosides are essential components of membrane microdomains that modulate the functioning of membrane proteins such as the insulin- or the epidermal growth factor receptor through a lateral association mechanism[19]. In addition, they have emerged as important regulators of metabolism, glucose homeostasis, and body weight[20], all functions that are highly relevant to GHSR signaling. To complete the picture of how the different properties of the lipid bilayer could regulate GPCRs signaling, we analyzed whether, besides selective lipid:protein interactions, the physical properties of the membrane, namely its thickness, also affected GHSR functioning. Taken together, our results illuminate a mechanism where the membrane could allosterically control ghrelin signaling by modulating its receptor conformational landscape.

## Results

**PIP2:GHSR interaction.** We first analyzed whether PIP2 interacted with the recombinant GHSR inserted into lipid nanodiscs. For the receptor to be surrounded with a significant number of lipid molecules, we used cNW30 that is an engineered variant of nanodisc scaffolding proteins designed to make large, covalently circularized, discs[21]. To ensure insertion of essentially a single GHSR protomer per disc, nanodisc assembly was carried out with the receptor immobilized on a solid matrix and a 10-fold molar excess in cNW30[7,22]. In addition, we used a large excess in lipids, as this favors proper formation of the large nanodiscs[21]. The GHSR-containing nanodiscs obtained under these conditions had an estimated stokes diameter in the 17 nm range (Supplementary Fig. 1a). Insertion into such large nanodiscs had no impact on GHSR pharmacological properties (Supplementary Fig. 1b,c; Supplementary Table 1). To analyze the effect of PIP2 on GHSR functioning, we then reconstituted GHSR into cNW30/POPC nanodiscs containing increasing amounts in this lipid. To this end, PIP2 was directly mixed with POPC before nanodisc assembly. As shown in Supplementary Fig. 2a with the fluorescent analog Bodipy-FL PIP2, the amount of PIP2 in the nanodiscs directly reflected its amount in the initial mixture.

We first investigated whether PIP2 molecules in the nanodisc interacted with the receptor. To this end, we used an assay initially developed with TREK-1 that relies on the FRET signal between a fluorescent analog of PIP2 (Bodipy-FL PIP2) and a fluorophore attached to the cytoplasmic region of the protein[23]. This assay has been shown to provide protein:PIP2 interaction profiles similar to those obtained with the unmodified lipid using native mass spectrometry (MS)[24]. The donor moiety, Lumi-4-Tb, was attached here to the cytoplasmic part of the GHSR sixth transmembrane (TM) domain through a unique reactive cysteine we had introduced at position $255^{6.27}$ of a minimal cysteine mutant (superscript numbers follow Ballesteros–Weinstein numbering[25]). This $C255^{6.27}$ mutant was shown to have pharmacological properties very similar to those of the wild-type receptor[26]. Nanodiscs were assembled with the labeled receptor and increasing amounts in either Bodipy-FL PIP2 or Bodipy-FL phosphatidic acid (PA) used as a control. As PI did not compete with PIP2 for binding GHSR (see below), we also used Bodipy-FL PI as a negative control in the FRET assay (Supplementary Fig. 3). Indeed, the Bodipy-FL phosphoinositide derivatives contain a peptide bond that is absent from BODIPY-FL PA (Supplementary Fig. 4), and this could have affected the insertion of the fluorescent derivatives into the bilayer, and thus the FRET signal. The amounts in fluorescent lipids in the initial lipid mixture ranged from 0 to 3% (labeled lipid-to-POPC molar ratio). This resulted in fluorescent lipid-to-GHSR molar ratios in the nanodiscs ranging from 0 to ca. 7 (Supplementary Table 2). As shown in Fig. 1a, the transfer signal between the Lumi-4-Tb donor and the Bodipy-FL acceptor rapidly increased with the amount of fluorescent PIP2 in the nanodiscs, reaching a maximum for a PIP2-to-GHSR molar ratio of about 4–5. This corresponds to ca. 2% PIP2 in the initial lipid mixture (Supplementary Table 2). Above this PIP2-to-GHSR ratio, the

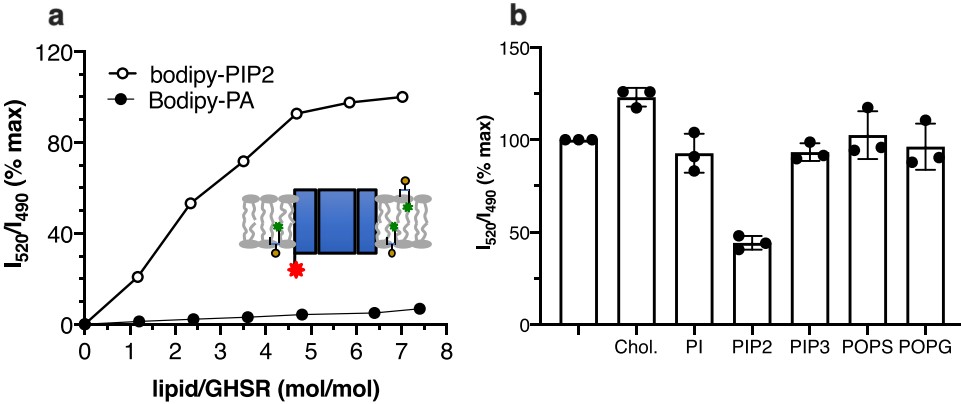

**Fig. 1 PIP2 binds to GHSR. a** FRET-monitored proximity assay with nanodiscs containing increasing amounts in Bodipy-FL PIP2 or Bodipy-FL PA and GHSR labeled with Lumi-4 Tb on C255[6,27]. **b** FRET-monitored competition between Bodipy-FL PIP2 (2.5 molar%) and unlabeled lipids. The latter were used at a limited concentration (2.5 molar%) not to affect the general physicochemical properties of the bilayer. The signal was normalized to that in the absence of competing lipids (first lane). Data in (**b**) are mean ± SD of three experiments. Source data are provided as a Source data file.

signal slightly increased with a slope similar to that of the control lipid Bodipy-FL PA. The reason for this slight increase is not clear at the present stage of the analysis, but it could be due to very transient interactions between BODIPY-FL PIP2 and low-affinity sites, as observed in the computational analyses (see below), or to non-specific effects associated with a crowding of the labeled lipids near the receptor because of the confined nature of the nanodisc structure. Taken together, this nevertheless indicates the occurrence of a limited number of specific PIP2-binding sites in GHSR. Combined with the experimental conditions used (0.5 μM receptor concentration), the FRET data are suggestive of an affinity for PIP2 in the sub-μM range for these sites.

This FRET assay has been shown to be well-adapted for monitoring the competition between labeled PIP2 and unlabeled lipids[24]. We thus used it to analyze whether other lipids could affect the proximity between GHSR and PIP2. The structure of the compounds used in this assay is given in Supplementary Fig. 4. As shown in Fig. 1b, unlabeled PIP2 significantly decreased the Lumi-4 Tb:Bodipy-FL transfer signal. As the acyl chain is different in both cases, this suggests that the selectivity of the interaction is mostly driven by the nature of the lipid headgroup. Accordingly, neither PI nor PIP3 decreased this signal (Fig. 1b). The absence of competition with PI was consistent with the fact that no significant FRET signal was observed when using Bodipy-FL PI instead of Bodipy-FL PIP2 (Supplementary Fig. 3). This was also the case for cholesterol and the negatively charged lipids POPS and POPG (Fig. 1b). The absence of competition with POPS or POPG is consistent with the fact that basic stretches in membrane proteins sequester multivalent lipids more effectively than monovalent ones[27]. Interestingly, cholesterol triggered a slight but significant increase in the FRET signal (Fig. 1b). This suggests this sterol could allosterically favor the distribution and/or change the arrangement of PIP2 molecules around GHSR, either because of a direct effect on the receptor or because of its impact on the physicochemical properties of the bilayer.

**PIP2-binding sites.** The significantly lower transfer efficiency between fluorescent PIP2 and GHSR we observed when the donor was located in the extracellular part of the receptor (C304[7.34]) suggested that PIP2-binding sites might be preferentially located in the cytoplasmic regions of the receptor (Supplementary Fig. 5). The absence of FRET signal in this case was not due to mislabeling of C304[7.34], as the absorption and emission spectra of the labeled receptor attested for the presence of Lumi-4 Tb (Supplementary

Fig. 5). To further identify the PIP2-binding sites, we developed a strategy combining molecular dynamics (MD) to site-directed mutagenesis. We first used MD to identify possible PIP2-binding sites on GHSR. To this end, we carried out a series of coarse-grained molecular dynamics (CGMD) simulations with GHSR embedded into the membrane system related to that we used in the FRET experiments, i.e., a POPC bilayer containing 7 PIP2 molecules on each side of the membrane. This number of PIP2 was based on the FRET data that indicates GHSR binds a maximum of 5–6 PIP2 (Fig. 1a). These simulations were performed on both the X-ray captured inactive form of GHSR (PDB 6KO5)[28] and on an active form we generated by homology with other receptors from the same family (see "Methods"). These CGMD simulations first confirmed that PIP2 molecules bind to the intracellular side of the receptor tighter than to its extracellular regions (Supplementary Fig. 6a), consistent with the FRET data in Supplementary Fig. 5. Besides, a finite number of PIP2 molecules bound at specific sites in GHSR was observed in the simulations as the result of an interaction between the negatively charged phosphorylated inositol headgroup of PIP2 and Lys/Arg residues in the cytoplasmic face of the GHSR transmembrane domains (Supplementary Fig. 6b, c). Specifically, three different sites were observed that bound a maximum of six PIP2 molecules (Fig. 2a–c, Supplementary Fig. 7), which is in the same range than the number of bound PIP2 estimated from the FRET-based assay. The first site (site 1) was formed by residues in the intracellular parts of TM1 and TM4 and bound a maximum of two PIP2 molecules (Supplementary Fig. 7). It included R70[1.59] and R72 at the TM1/ICL1 junction and K157[4.39] and K161[4.43] in TM4 (Fig. 2a). Residues 1.59, 4.39, and 4.43 had also been proposed to participate to PIP2 binding to NTS1R[9] and A2AR[17]. The second site (site 2) was responsible for binding 1 to 2 PIP2 molecules (Supplementary Fig. 7); it included R237[5.63], R242[5.68], R243[5.69], and R244[5.70] in TM5 as well as K259[6.31] in TM6 (Fig. 2b). This second site is slightly different from that proposed for NTS1R, which is mostly composed of residues from TM4[9], but similar to that found in A2AR[17]. Finally, the last one (site 3) bound 1 PIP2 molecule (Supplementary Fig. 7); it was formed by K328[8.48] and R331[8.51] at the TM7/H8 interface (Fig. 2c), a site that was also found in NTS1R[9] and A2AR[17]. These three sites were systematically retrieved in all three independent simulations. Other less specific sites could nevertheless be occasionally observed in the simulations that could explain the additional slight increase in the FRET signal observed at high PIP2-to-receptor molar ratios.

We then performed site-directed mutagenesis on the GHSR minimal cysteine mutant, replacing the putative PIP2-binding

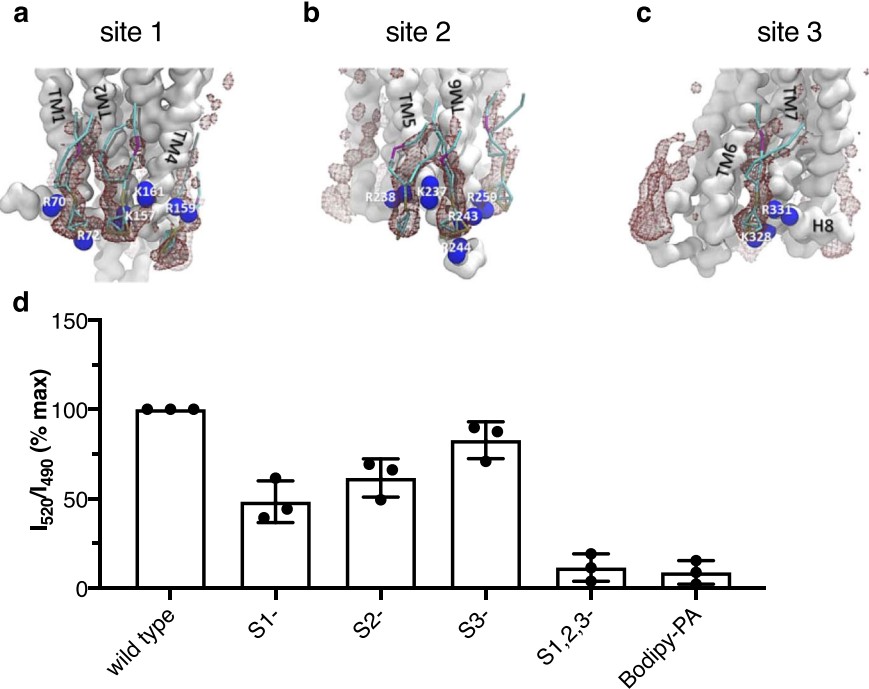

**Fig. 2 PIP2-binding sites. a–c** Representative snapshots showing the position of the lipid(s) in the three sites (active conformation) that were most occupied by PIP2 (**a** site 1; **b** site 2; **c** site 3). The residues interacting with the lipid within each of these sites are indicated (blue), the PIP2 molecules are reported in sticks colored according to the CG beads and the receptor backbone (active conformation) as a white surface. **d** FRET signal between Bodipy-FL PIP2 (2.5 molar%) and Lumi-4 Tb attached to C255[6.27] of the wild-type receptor or of the putative PIP2-binding sites mutants. The FRET signal obtained with negative control labeled lipid Bodipy-PA is given for comparison. The signal was normalized to that measured with wild-type GHSR. Data in (**d**) are mean ± SD of three experiments and is provided as a Source data file.

residues in either site 1 (S1 mutant), site 2 (S2 mutant), site 3 (S3 mutant), or in all three sites (S1,2,3 mutant) with alanines. Such replacements did not significantly alter the pharmacological profile of GHSR in the absence of PIP2 (Supplementary Fig. 8). This indicates that the residues we mutated, all located at the lipid:protein interface, directly participate neither to the fold of the receptor nor to the interaction with the ligand and/or the G protein. Each mutant receptor was then labeled with the Lumi-4 Tb donor on C255[6.27], and the FRET-based assay used to monitor the interaction with fluorescent PIP2. A significant attenuation of the FRET signal was observed with each of the mutants (Fig. 2d). Specifically, mutating sites 1, 2, and 3 led to a ca. 50%, 40%, and 20% decrease in the signal, respectively. This distribution is consistent with the number of PIP2 molecules interacting with each site in the CGMD analyses (Supplementary Fig. 7). Perhaps not surprisingly, the signal obtained with the S1,2,3 mutant was essentially similar to that measured with the negative control Bodipy-PA, suggesting that mutating all three sites abolished specific PIP2 binding to GHSR. Taken together, this indicates that the three sites we identified in the CGMD simulations are hotspots for PIP2 binding to GHSR.

**PIP2 binding affects GHSR conformation**. To delineate how PIP2 binding affected GHSR functioning, we first explored the impact of this lipid on the conformational features of the isolated receptor using the fluorescence properties of monobromobimane (MB) attached to C255[6.27], a position that has been extensively used with the β2AR[29]. When MB is located at this position, changes in its fluorescence properties primarily report on the movements of TM6 associated with GPCR activation[29]. Figure 3a shows the emission spectrum of MB-labeled GHSR. This profile reflects the average, equilibrium signal between the different receptor conformational states present in the solution. Hence, any

change in the distribution of these states should modify the fluorescence emission spectrum. Accordingly, ghrelin binding was associated with a significant change in the MB emission properties, with a decrease in the emission intensity and a concomitant red-shift in the maximum emission wavelength $\lambda_{max}$, whereas binding of the inverse agonist LEAP2(1-14) resulted in an opposite change when compared to the apo receptor (Fig. 3a, b). Incorporation of 2.5% PIP2 into the lipid nanodiscs resulted in an additional decrease in the emission intensity and increase in $\lambda_{max}$ for both the apo and ghrelin-loaded GHSR. In contrast, no significant effect was observed for LEAP2-loaded GHSR (Fig. 3a, b), suggesting that the effects of PIP2 were somehow related to the active state of the receptor (see below). These effects were not due to the possible insertion of the ghrelin acyl moiety into the bilayer, as the same behavior was observed with MK0677 and JMV1843, two non-peptide GHSR full agonists (Supplementary Fig. 9). The effect was specific for PIP2, as neither PIP3 nor POPG affected the MB emission spectrum in a significant manner (Supplementary Fig. 10). In contrast, no effect of PIP2 on the MB emission profile was observed with the S1,2,3 mutant (Fig. 3c), indicating that the effect of PIP2 is due to its direct interaction with the receptor rather than to an effect on the bilayer properties. As for β2AR[29], the G protein further stabilized the active state of GHSR, as evidenced by the additional change in the emission properties of MB upon adding Gq to the ghrelin-loaded receptor (Fig. 3a, b). In this case also, the presence of PIP2 in the nanodiscs was associated with an additional decrease in the emission intensity and increase in $\lambda_{max}$ (Fig. 3a, b). Taken together, this data suggests that PIP2 allosterically shifts the equilibrium away from the inactive state of GHSR.

The emission properties of MB essentially depend on its very local environment. Hence, it could not be excluded that similar changes in the probe fluorescence would nevertheless result from

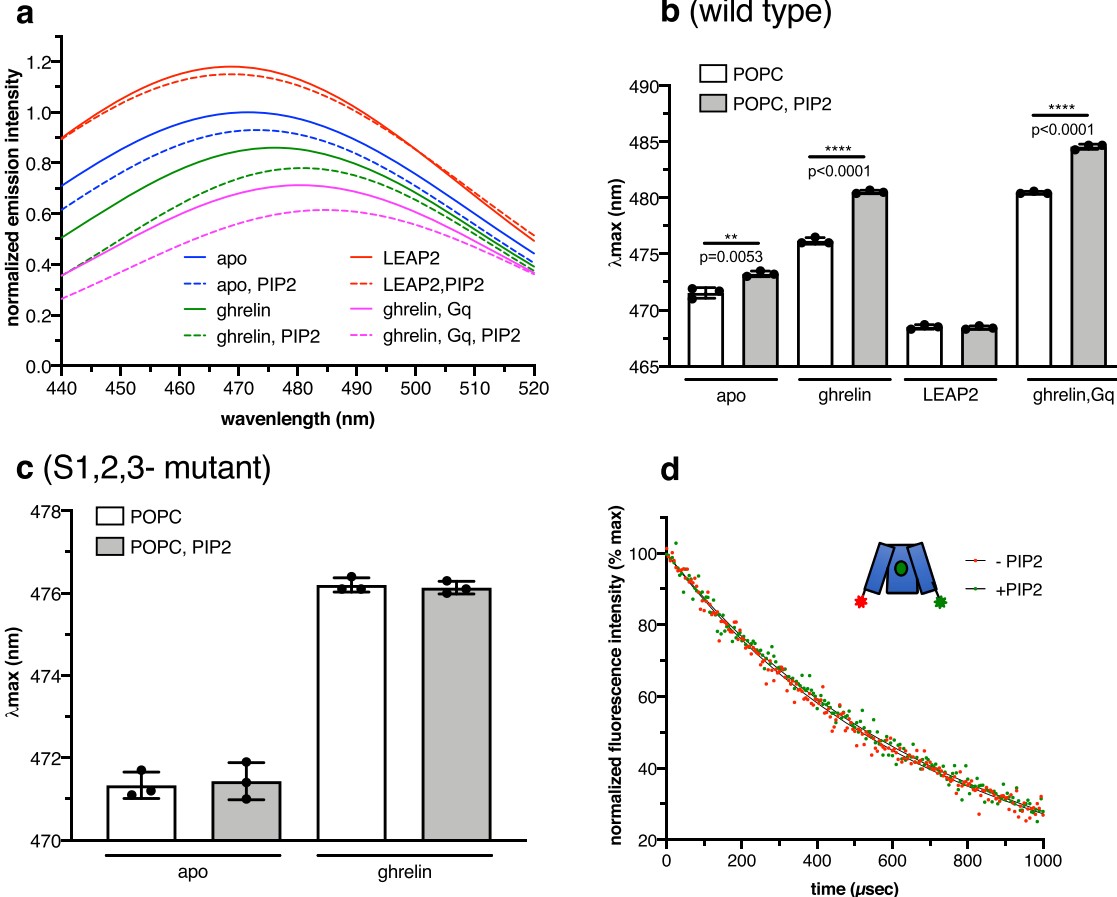

**Fig. 3 PIP2 modulates GHSR conformation. a** MB emission spectra of GHSR in POPC or POPC:PIP2 (2.5% PIP2) nanodiscs in the absence of ligand, in the presence of 10 μM ghrelin, in the presence of 10 μM LEAP2(1-14) or in the presence of 10 μM ghrelin and the Gα$_q$β$_1$γ$_2$ trimer (1:5 receptor-to-G protein molar ratio). **b, c** Changes in $\lambda_{max}$ for the wild-type receptor and the mutants of the PIP2-binding sites. Data in (**b**) and (**c**) are mean ± SD of three experiments. Statistical values were obtained by means of unpaired Student's t test **0.001 < p < 0.01, ***0.0001 < p < 0.001). **d** Intramolecular sensitized-emission decays from isolated ghrelin-loaded GHSR (10 μM ligand) in the absence of G protein and in the absence or the presence of PIP2 (2.5% molar ratio), with the donor and acceptor fluorophores in the cytoplasmic end of TM6 and TM1. Data are presented as normalized fluorescence intensity as a function of time and represent the average of three measurements. Source data are provided as a Source data file.

different changes in GHSR conformation. To further illuminate the impact of PIP2 on the distribution of GHSR conformational states, we carried out an additional series of intramolecular LRET measurements with a Lumi-4 Tb donor attached to C255[6.27] and an Alexa Fluor 488 acceptor bound to the unnatural amino-acid p-azido-L-phenylalanine (AzF) at position 71[1.60] in the cytoplasmic tip of TM1[26]. LRET has several technical advantages over conventional FRET, including distance measure with greater accuracy, and insensitivity to incomplete labeling[30]. Indeed, in LRET, measurements are usually carried out via sensitized-emission lifetime measurements, and thus the donor-only and acceptor-only species do not contribute to the signal[30]. Hence, in a multi-exponential decay analysis of the lifetime of the sensitized emission of acceptor, the pre-exponential terms are directly related to the distance between the probes and to their populations, therefore providing an estimation of both parameters[30]. To be noted, although it cannot be excluded that the size of the Lumi-4 probe may affect the absolute values of the receptor dynamics, we nevertheless previously showed that LRET was well-adapted to monitor the changes in GHSR conformation in a variety of conditions[26]. Indeed, because of the location of the two probes, they directly report on the amplitude of the TM6 outward movements, a hallmark of GPCR activation[26].

Accordingly, when inserted at a closely related position in TM6 of the vasopressin receptor V2R, Lumi-4 also consistently reported the receptor dynamics[31]. We thus recorded here the acceptor-sensitized-emission profiles of the ghrelin-loaded labeled GHSR in POPC nanodiscs containing or not 2.5% PIP2 (Fig. 3d). In both cases, the acceptor-sensitized emission decays were best described by a double exponential function. The decay time of the two components inferred from these exponentials was similar whether PIP2 was present or not (Table 1). The only difference was the population of the slow component, which we previously assigned to the active/active-like GHSR conformation[26], which increased in the presence of PIP2 (Table 1). This difference in population would correspond to a PIP2-dependent decrease in the overall free energy difference between the inactive and active/active-like states of GHSR of about 0.4 kcal/mol. Of importance, even such a relatively modest energetic contribution might yield a detectable shift in signaling, based on a simple thermodynamic model[32]. Taken together, this data indicates that PIP2 does not modify the geometrical features of the receptor, at least as far as the relative orientation of TM1 and TM6 is considered, but rather affects the relative populations of the different states in the receptor conformational repertoire, favoring the active/active-like one.

**Table 1 Lifetimes of Alexa Fluor 488 sensitized emission and corresponding molecular fractions.**

| Species | $\tau_{ad1}$ ($\mu$s) | A1 (%) | $\tau_{ad2}$ ($\mu$s) | A2 (%) |
|---|---|---|---|---|
| GHSR/ghrelin/POPC | 294.0 ± 7.2 | 35.3 ± 0.7 | 826.8 ± 13.8 | 64.7 ± 0.3 |
| GHSR/ghrelin/POPC + PIP2 | 289.4 ± 8.4 | 21.8 ± 0.3 | 815.7 ± 14.7 | 78.2 ± 1.1 |
| GHSR/DMoPC | 267.2 ± 18.7 | 63.7 ± 0.8 | 557.8 ± 12.2 | 36.3 ± 0.9 |
| GHSR/DOPC | 275.9 ± 11.5 | 64.2 ± 0.9 | 854.7 ± 17.9 | 35.8 ± 0.2 |
| GHSR/ghrelin/DMoPC | 281.7 ± 8.2 | 39.1 ± 0.5 | 547.2 ± 14.3 | 60.9 ± 0.8 |
| GHSR/ghrelin/DOPC | 286.3 ± 9.6 | 34.8 ± 0.2 | 834.6 ± 17.9 | 65.2 ± 0.3 |
| GHSR/ghrelin/POPC/Gq | 599.3 ± 9.0 | 25.0 ± 0.8 | 1070.2 ± 9.9 | 75.0 ± 1.0 |
| GHSR/ghrelin/POPC + PIP2/Gq | 617.2 ± 13.5 | 11.7 ± 1.1 | 1062.0 ± 11.3 | 88.3 ± 1.3 |
| GHSR/ghrelin/POPC/Gi2 | 891.7 ± 14.1 | 4.3 ± 0.3 | 1074.6 ± 18.9 | 95.7 ± 0.2 |
| GHSR/ghrelin/POPC + PIP2/Gi2 | 881.2 ± 15.7 | 2.7 ± 0.5 | 1049.3 ± 19.3 | 97.3 ± 0.9 |
| GHSR/ghrelin/DMoPC/Gq | 613.6 ± 16.3 | 92.5 ± 0.8 | 1074.7 ± 7.7 | 7.5 ± 0.5 |
| GHSR/ghrelin/DOPC/Gq | 595.2 ± 14.9 | 24.8 ± 1.2 | 1069.4 ± 11.7 | 75.2 ± 1.3 |
| GHSR/ghrelin/DMoPC/Gi2 | 918.4 ± 6.8 | 96.4 ± 1.0 | 1026.3 ± 13.3 | 3.6 ± 1.2 |
| GHSR/ghrelin/DOPC/Gi2 | 898.1 ± 12.1 | 3.2 ± 0.9 | 1029.8 ± 8.1 | 96.8 ± 0.7 |

The sensitized emission $\tau_{ad1}$ and $\tau_{ad2}$ were calculated from the two dominant exponential components of sensitized emission decays. The molecular fractions A1 and A2, expressed in % of the total population, were calculated from the pre-exponential factors and the excited-state lifetime values[66]. The species considered are schematically presented (receptor in blue, ligand in green, G protein in yellow).

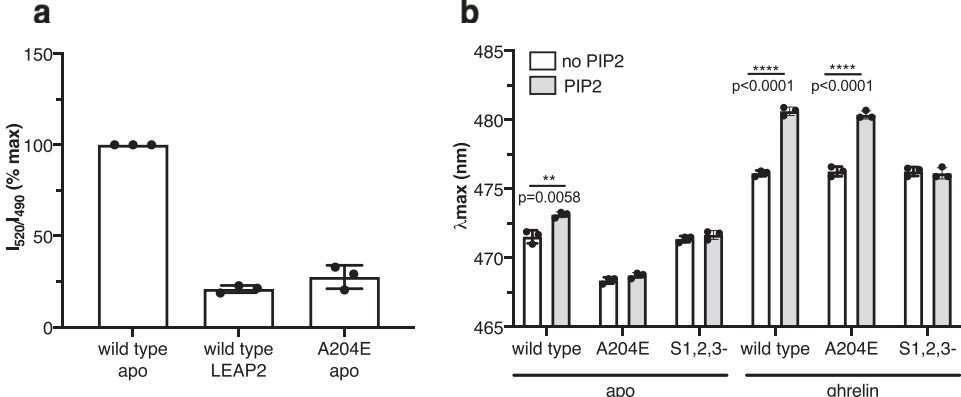

**Fig. 4 PIP2 binds preferentially to the active state of GHSR. a** FRET signal between Bodipy-FL PIP2 (2.5 molar%) and Lumi-4 Tb attached to C255[6.27] of the wild-type receptor or of the GHSR-A204E mutant. The signal obtained with the S1,2,3 mutant is given for comparison. The signal was normalized to that measured with wild-type GHSR. **b** Changes in MB emission $\lambda_{max}$ for wild-type GHSR, the A204E, and the S1,2,3 mutant in the absence or in the presence of 10 μM ghrelin. Data are mean ± SD of three experiments. Statistical values were obtained by means of unpaired Student's t test (**$0.001 < p < 0.01$, ****$p < 0.0001$). Source data are provided as a Source data file.

**PIP2 preferentially binds to the active state of GHSR.** We then investigated the mechanism responsible for the effect of PIP2 binding on the distribution of the different GHSR conformational states. PIP2 had very little impact on the MB emission properties of the wild-type GHSR bound to the inverse agonist LEAP2(1-14) (Fig. 3a, b). Moreover, a significant decrease in the FRET signal between labeled GHSR and BODIPY-FL PIP2 was observed in this case (Fig. 4a). Hence, a possibility would be that the effect of PIP2 resulted from a preferential binding to the active state of GHSR, thereby stabilizing it. In this model, the significant proportion of PIP2 binding to apo GHSR would result from the occurrence of the significant proportion of active state associated with the receptor high constitutive activity[26]. To further assess this mechanism on an experimental basis, we analyzed the effect of PIP2 on the A204E mutant of GHSR (GHSR-A204E). This natural mutant displays no constitutive activity[14,33] because its landscape is mostly populated with one, inactive, conformation, but can be fully activated by ghrelin[26]. As shown in Fig. 4a, we observed a significant decrease in the FRET signal between fluorescent PIP2 and apo GHSR-A204E, in the same range than that observed for the LEAP2-loaded wild-type GHSR. Moreover,

PIP2 had essentially no effect on MB fluorescence attached to C255[6.27] of the apo GHSR-A204E (Fig. 4b). Taken together, these data indicate that PIP2 likely interacts preferentially with the active state of GHSR, thereby explaining its impact on the equilibrium between the inactive and active/active-like states of the receptor.

The conclusion that PIP2 binding stabilizes the active conformation of GHSR was supported by CG simulations performed on both inactive and active conformations of GHSR (Supplementary Fig. 11). To be noted, PIP2 was still bound to the inactive state of GHSR in these simulations, in contrast to the FRET experiments where a significant decrease in the signal was observed for the apo A204E mutant and the LEAP2-loaded wild-type receptor. A possible explanation for this discrepancy would be that the decreased affinity of PIP2 for its binding sites in the receptor inactive state invoked to explain the lower FRET signal (see above) would not be observed in the simulations because of the particular features of CGMD that make the lipids stick to the protein[34], and thus do not allow any change in the protein:lipid exchange dynamics to be visualized. Interestingly, we nevertheless observed a significant redistribution of PIP2 in the active

conformation within site 3. Indeed, PIP2 was able to intercalate between TM6 and TM7/H8 in this state, whereas this position was clearly not observed in the inactive conformation (Supplementary Fig. 11). This difference can be also observed in Supplementary Fig. 7 that shows a significant decrease in the percentage of frames displaying 1 bound PIP2 in site 3, from more than 60% in the active conformation to <40% in the inactive one. As the spreading of TM5/TM6 from TM7/H8 is one of the major events associated with GPCR activation, this could be a possible mechanism to explain how PIP2 at this position stabilizes the active/open conformation of GHSR.

**PIP2 binding affects GHSR-catalyzed G protein activation.** We then analyzed whether the effect of PIP2 binding to GHSR conformation reflected in its ability to activate two of the main G protein subtypes it couples to, Gq and Gi2[22]. GHSR also couples to G13[13,14] but coupling to this G protein subtype could not be analyzed here, as we could not get purified G13 for the GTP turnover assays. In the absence of PIP2, apo GHSR triggered a significant GTP turnover for Gq only (Fig. 5a, b), in agreement with its constitutive activity in the Gq pathway[14]. Ghrelin then increased GTP turnover for both Gq and Gi2 whereas LEAP2 reduced basal GTP binding to Gq (Fig. 5a, b)[35]. The presence of 2.5% PIP2 in the lipid nanodiscs further increased GTP turnover for Gq and Gi2 (Fig. 5a, b). This effect was specific for PIP2, as neither the negatively charged lipid POPG nor PIP3 affected GTP turnover to a similar extent when added in similar amounts into the nanodiscs (Supplementary Fig. 12). It was not due to an interaction of the acyl moiety of ghrelin with the lipid bilayer either, as the same effect was observed with MK0677 and JMV1843 (Supplementary Fig. 9). In agreement with the FRET-based assay, increasing the PIP2 amount in the nanodisc up to a 5% molar ratio did not further affect GTP turnover, indicating a saturable effect, likely resulting from a specific binding to GHSR. This was confirmed by the absence of effect of PIP2 on either basal or ghrelin-induced GTP turnover when using the S1,2,3 mutant instead of the wild-type receptor (Fig. 5c, d). Taken together, these results indicated that binding of PIP2 to GHSR increases its efficacy at activating Gq and Gi2 proteins.

We finally analyzed whether this increase in GTP turnover resulted from a different GHSR:G protein mode of interaction or simply from the increase in the population of GHSR active state. To this end, we monitored the intermolecular LRET profiles between the receptor and the G protein α subunit[26]. In this case, the Lumi-4 Tb donor was attached to the N-terminus of the Gα subunit through a reaction with the NHS derivative of the probe whereas the Alexa Fluor 488 acceptor was attached to the cytoplasmic part of TM1 through a copper-free click chemistry reaction with the AzF residue at position 71[1.60]. We previously demonstrated that the resulting LRET profile directly reflects the geometrical arrangement of the receptor:G protein complex[26]. The LRET profiles with Gq were best fitted with a two exponential with a slow and a fast component (Fig. 5e). These components had been assigned to the inactive preassembled and the active GHSR:Gq complexes, respectively[26]. The decay times were comparable whether PIP2 was present or not in the nanodiscs (Table 1), suggesting a similar arrangement of the receptor:G protein assembly. As in the case of the intramolecular LRET, only the proportion of the slow component corresponding to the active complex increased upon adding PIP2 to the nanodiscs (Table 1). Based on the variation in these populations, it could be concluded that PIP2 energetically favored the formation of the active GHSR:Gq complex (ΔΔG of about −0.5 kcal.mol⁻¹, as calculated from the populations in Table 1). In the same way, a single very major species was observed for Gi2

whether PIP2 was present or not (Fig. 5e; Table 1). This species certainly corresponded to the active complex, while GHSR does not preassemble with this G protein subtype[26]. Taken together, these data indicate that PIP2 increases the population of the receptor:G protein active complex, possibly because of the increase in the population of receptor active state, without significantly affecting the geometrical features of this assembly.

**GM3 binds GHSR and affects receptor conformation and G protein activation.** We then analyzed if other lipids besides PIP2 could interact with GHSR. To this end, GHSR-containing nanodiscs were assembled with 5% GM3 (total lipid-to-GM3 molar ratio), which is in the range of the amounts of sphingolipids found in eukaryotic plasma membranes[19]. Incorporation of GM3 into the nanodiscs was carried out as described above for PIP2. To be noted, like with PIP2, the nanodisc system does not allow to preserve the asymmetry of the plasma membrane where GM3 is only located in the extracellular leaflet of the bilayer[19]. To first assess whether GM3 interacted with GHSR, we used the FRET-based assay with a fluorescent derivative of this lipid (commercial name TopFluor GM3) whose spectral characteristics are similar to those of the Bodipy-FL modified lipids used above. The Lumi-4 Tb donor was bound either to C255[6.27] in the intracellular part of GHSR or to C304[7.34] in its extracellular region. As shown in Fig. 6a, a significant FRET signal was observed in both cases, suggesting that GM3 interacted with GHSR at its extra- and intracellular sides. Interestingly, PIP2 (but not POPG) abolished the FRET signal with the probe at position C255[6.27] (Fig. 6a). Moreover, no significant FRET signal was observed with the Lumi-4 Tb probe at position C255[6.27] when using the S1,2,3 mutant instead of the wild-type receptor (Fig. 6a). In contrast, neither PIP2 nor POPG had a significant effect on the FRET signal between Lumi-4 Tb at position C304[7.34] and TopFluor GM3, and mutating the PIP2-binding sites did not significantly alter this signal (Fig. 6a). Taken together, these data suggest that GM3 interacts with intra- and extracellular sites of GHSR isolated in nanodiscs, with PIP2 and GM3 likely sharing the same cytoplasmic sites whereas the extracellular sites would be selective of GM3.

In order to further assess how GM3 bound to GHSR, CGMD simulations were performed under the conditions described above but where the PIP2 molecules were replaced by GM3. These simulations confirmed that GM3 indifferently bound to the intra- and the extracellular sides of GHSR (Supplementary Fig. 13). This resulted from a specific interaction, as no binding of POPG was observed at the same threshold using the same number of lipids. As shown in Supplementary Fig. 13, GM3 bound the same intracellular sites than those identified for PIP2, suggesting that both lipids could compete for binding these sites, consistent with the FRET data described above. To confirm this competition, we performed additional simulations in which PIP2 was mixed with either GM3 or POPG molecules. As a result, we observed a significant redistribution of PIP2 molecules around the receptor in the presence of GM3 (Supplementary Fig. 13d). Such a competition was clearly identified in Supplementary Fig. 14 that shows a decrease in the number of PIP2 molecules bound to the intracellular sites of GHSR in presence of GM3, while POPG had no significant effect.

We then explored the impact of GM3 on the conformational features of GHSR using MB fluorescence. As shown in Fig. 6b, incorporation of 5% GM3 into the lipid nanodiscs increased MB emission $\lambda_{max}$ compared to nanodiscs formed of POPC only, suggesting that this lipid further stabilized the active state of the receptor. This effect was observed in the absence and in the presence of ghrelin. For the apo receptor, the increase in MB

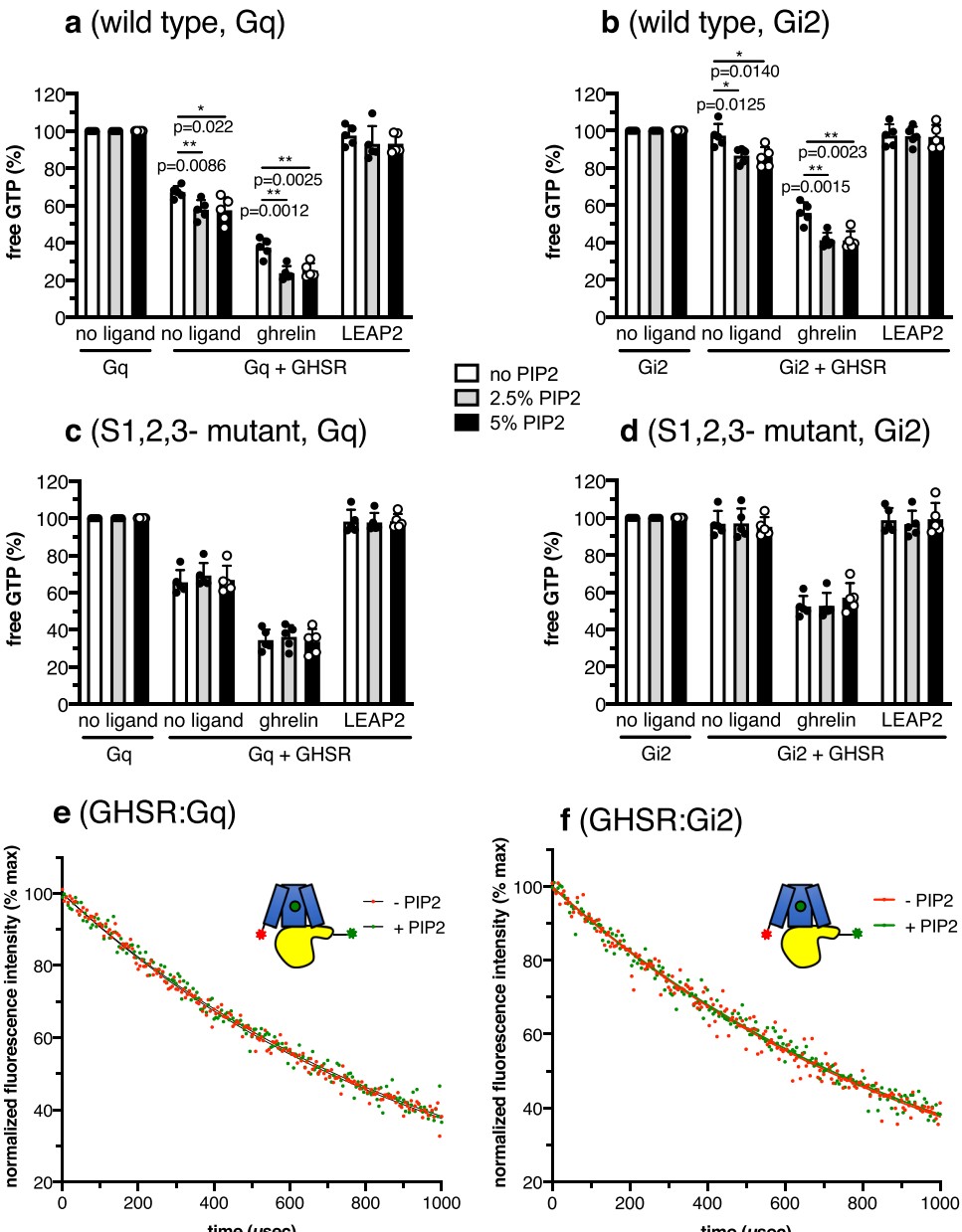

**Fig. 5 PIP2 modulates G protein activation.** GTP turnover for Gq (**a**, **c**) and Gi2 (**b**, **d**) catalyzed by wild-type GHSR (**a**, **b**) or its PIP2-binding sites mutant (**c**, **d**) in nanodiscs containing different amounts in PIP2 in the absence of ligand, in the presence of 10 μM ghrelin or in the presence of 10 μM LEAP2(1-14). In all cases, the signal was normalized to that obtained for the G protein in the absence of receptor, and data are mean ± SD of five experiments. Statistical values were obtained by means of unpaired Student's $t$ test (*$0.01 < p < 0.05$, **$0.001 < p < 0.01$). **e**, **f** Intermolecular sensitized-emission decays from $G\alpha_q\beta_1\gamma_2$ (**e**) or $G\alpha_{i2}\beta_1\gamma_2$ (**f**) and ghrelin-loaded GHSR (10 μM ligand) assembled into nanodiscs containing or not PIP2 (2.5% molar ratio), with the Lumi-4 Tb donor on the Gα N-terminus and the Alexa Fluor 488 acceptor on the cytoplasmic end of GHSR TM1. In each case, data are presented as normalized fluorescence intensity as a function of time and represent the average of three measurements. Source data are provided as a Source data file.

emission $\lambda_{max}$ observed with GM3 was in the same range than that observed with PIP2, and the presence of both GM3 and PIP2 in the nanodiscs did not further change the $\lambda_{max}$ value compared to PIP2 alone (Fig. 6b). In contrast, for the ghrelin-loaded receptor, incorporation of both PIP2 and GM3 in the nanodiscs triggered a larger increase in $\lambda_{max}$ compared to PIP2 alone (Fig. 6b), suggesting that both lipids could cooperate to stabilize the agonist-dependent active state. As PIP2 and GM3 share the same intracellular binding site but not the extracellular ones, a possible model would be that binding of lipids to intracellular and extracellular sites affects in a distinct manner the basal and ligand-dependent GHSR conformational states. To further illuminate this possible mechanism, we

analyzed the impact of GM3 on the emission properties of MB bound to C255[6.27] of the S1,2,3 mutant. This mutant binds GM3 at its extracellular sites only. As such, it mimics to some extent the asymmetry of the natural plasma membrane with regard to the interactions with GM3. As shown in Fig. 6b, binding of GM3 to the extracellular sites of the GHSR S1,2,3 mutant did not significantly affected the MB emission properties for the apo receptor. In contrast, in the presence of ghrelin, GM3 triggered a significant change in the MB emission $\lambda_{max}$ for the S1,2,3 mutant. This was not due to an interaction of the acyl moiety of ghrelin with GM3 or with the lipid bilayer, as the same effect was observed with the non-peptide agonists MK0677 and JMV1843 (Supplementary Fig. 9a).

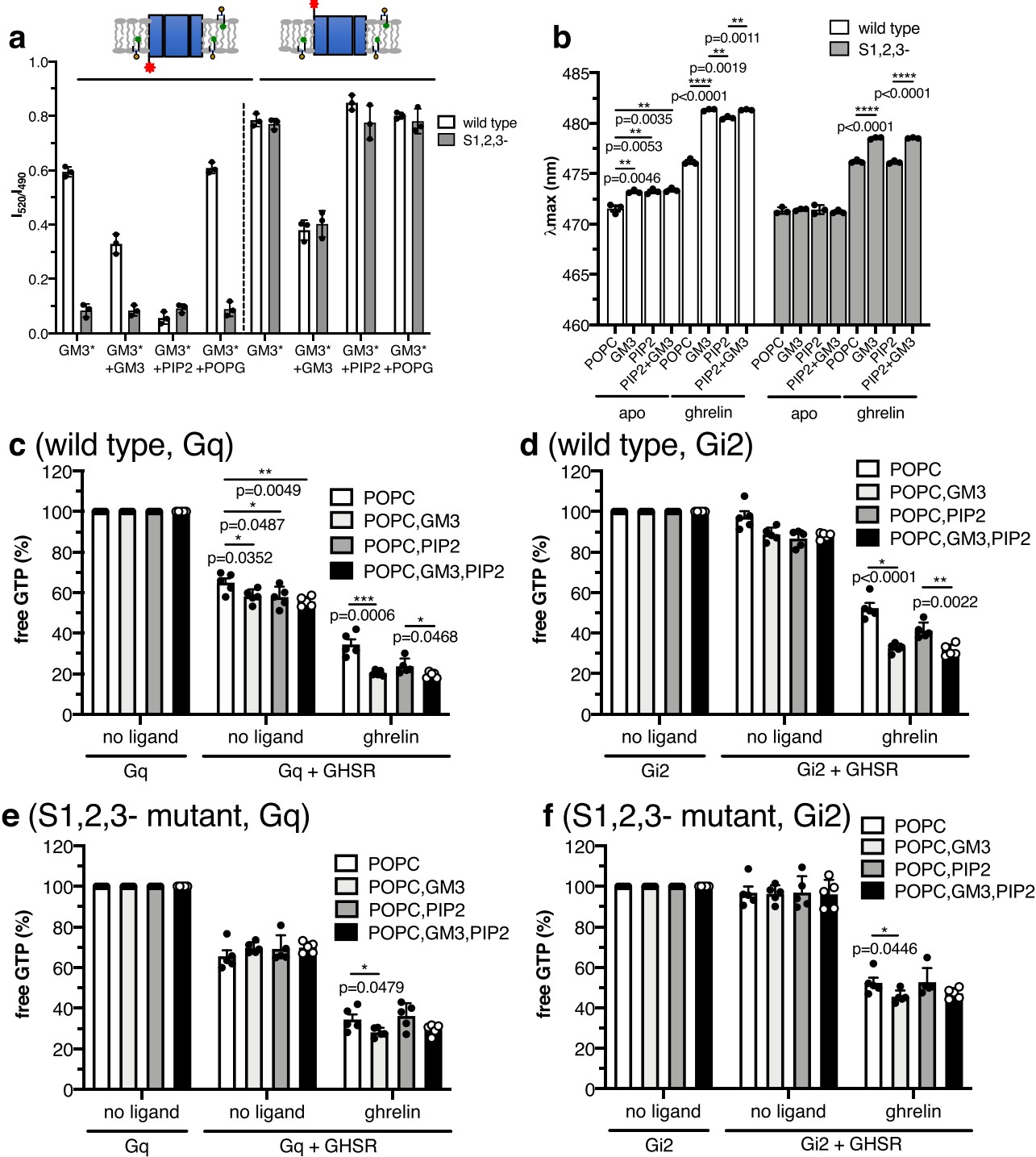

**Fig. 6 GM3 binds GHSR and modulates G protein activation. a** FRET signal between C11 TopFluor GM3 (GM3*; 5 molar%) and Lumi-4 Tb attached to either C255[6.27] or C304[7.34] of GHSR in the absence of competing lipid or in the presence of 5% (molar ratio) unlabeled GM3, PIP2, or POPG. **b** Changes in MB emission $\lambda_{max}$ for the wild-type receptor and the PIP2-binding sites mutant assembled into nanodiscs containing 5% GM3, PIP2 or GM3 and PIP2 in equimolar amounts in the absence or presence of 10 μM ghrelin. Data are mean ± SD of three experiments. **c–e** GTP turnover for Gq and Gi2 catalyzed by wild-type GHSR (**c**, **d**) or its PIP2-binding sites mutant (**e**, **f**) in nanodiscs containing 5% GM3, PIP2 or GM3 and PIP2 in equimolar amounts, in the absence or presence of 10 μM ghrelin. The signal was normalized to that obtained for the G protein in the absence of receptor, and data are mean ± SD of five experiments. In all cases, statistical values were obtained by means of unpaired Student's $t$ test (*$0.01 < p < 0.05$, **$0.001 < p < 0.01$, ***$0.0001 < p < 0.001$, ****$p < 0.0001$). Source data are provided as a Source data file.

Taken together, these observations point to a model where GM3 binding to its extracellular sites impacts on agonist-dependent GHSR activation only.

We finally analyzed whether the effect of GM3 on GHSR conformation reflected in the ability of the receptor to activate G proteins. As shown in Fig. 6c, d, the presence of GM3 in the nanodiscs increased GTP turnover for Gq and Gi2, both in the absence of ligand (Gq) and in the presence of ghrelin (Gq, Gi2). With regard to basal Gq activation, the effect of GM3 was similar to that triggered by PIP2, and no further change was observed when adding both lipids to the nanodiscs (Fig. 6c). This could indicate that the impact of GM3 on GHSR constitutive activity essentially results from its binding to the PIP2 intracellular sites. Accordingly, GM3 had no effect on Gq basal activity for the S1,2,3 mutant (Fig. 6e). In contrast, in the presence of ghrelin, GTP turnover was statistically larger in the presence of both GM3 and PIP2 compared to PIP2 alone (Fig. 6c, d). This suggests that binding of GM3 to the extracellular sites could contribute to increase agonist-dependent G protein activation. Consistent with this model, the presence of GM3 also increased ghrelin-dependent GTP turnover for the S1,2,3 mutant (Fig. 6e, f). Taken together, these data suggest that the two lipids considered here have different effects on G protein activation. Specifically, binding of GM3 to the extracellular regions of GHSR would contribute to agonist-dependent GTP turnover whereas PIP2 binding to the intracellular sites would affect both agonist-independent and agonist-dependent G protein activation.

**The membrane thickness affects GHSR conformation**. In addition to specific lipid:protein interactions, the bulk properties of the bilayer have also been shown to modulate membrane protein conformational dynamics[5]. We thus subsequently addressed this process to illuminate the different ways in which the membrane could regulate GHSR signaling. Specifically, we focused on the effects of thickness of the bilayer, as this parameter can be easily controlled with the nanodisc model systems, which is not the case for other features like curvature or lateral pressure. The thickness of unsaturated PC bilayers has a linear dependence on the acyl chain length[36]. cNW30 nanodiscs were thus assembled with either DMoPC (14:1 (Δ9-Cis) PC) or DOPC (18:1 (Δ9-Cis) PC) (see Supplementary Fig. 4 for the structure of these lipids). PC is among the most common glycerophospholipids of higher eukaryotes and, although it is not the most abundant lipid, C14 PC nevertheless represents about 6% of the lipids extracted from mammal HEK cells membranes[37]. C14 PC was thus selected here as the lower limit length for a membrane phospholipid acyl chain. Besides, unsaturated acyl chains were used because the Tm of the corresponding phospholipids are significantly lower than the temperature at which all experiments were carried out, which is not the case of their saturated counterparts. The fluidity of the membrane was the same in both cases, as shown by the similar value obtained for the general polarization of the fluorophore laurdan inserted in DMoPC and DOPC nanodiscs (Supplementary Table 3). Hence, the two kind of nanodiscs should differ essentially in the thickness of their bilayer only, with a difference of 5–6 Å[36]. Changing the lipid composition on the nanodiscs did not dramatically affect the ligand-binding properties of GHSR in the absence of G proteins. Specifically, assembly into DOPC nanodiscs was associated with a slight increase in the affinity of GHSR for ghrelin and a concomitant decrease in its affinity for LEAP2(1-14) (Supplementary Fig. 15).

We first investigated the conformational features of the isolated GHSR in DMoPC and DOPC nanodiscs using the LRET signal between the Lumi-4 Tb donor and the Alexa Fluor 488 acceptor attached to C255[6.27] and AzF71[1.60], respectively. In the absence of

ligand, the acceptor-sensitized emission decays of the isolated receptor were again best described by a double exponential function (Fig. 7a). The fast component was in the same range independently of the lipid, indicating no effect of the bilayer thickness on the receptor inactive state (Table 1). In contrast, the slow component, which we previously assigned to the active/active-like state of GHSR[26], was significantly different (Table 1). Although a difference due to distinct exchange rates between the two states cannot be excluded at the present stage, this difference in the LRET efficiency might be indicative of a different distance between the two probes, as previously concluded from MD simulations[26]. Based on the decay time of the slow component and the spectroscopic features of the probes[26], the distance between the probes in the active/active-like state of GHSR might be in the 35 Å (DMoPC) and 40 Å (DOPC) range. This suggests that changing the membrane thickness of the bilayer had an impact on the conformational features of the active/active-like conformation of the isolated receptor in the absence of G protein. Specifically, the thinner the membrane the shorter the distance between the cytoplasmic ends of TM1 and TM6, with a difference in the 5 Å range. The same trend was observed in the presence of ghrelin (Fig. 7b). Indeed, agonist binding was associated only with a change in the relative distribution of the two components without significantly affecting the decay times (Table 1).

**The membrane thickness affects GHSR-catalyzed G protein activation**. To assess whether these differences in GHSR conformation had an impact on G protein activation, we measured receptor-catalyzed GTP turnover. As shown in Fig. 7c, essentially no GTP turnover was observed for Gq when GHSR was assembled into DMoPC nanodiscs. Of interest, both basal and ghrelin-induced GTP turnover were abolished. Consistent with this result, a tentative GDP-release assay suggested that no GDP was released from Gαq when GHSR was inserted into DMoPC nanodiscs (Supplementary Fig. 16). In contrast, DOPC nanodiscs preserved the Gq activation properties of GHSR, as a significant GTP turnover was observed both in the absence and in the presence of ghrelin (Fig. 7c). In contrast to Gq, modifying the membrane thickness did not alter Gi2 activation in a major manner (Fig. 7d). Indeed, receptor-catalyzed GTP turnover was observed for Gi2 in the presence of ghrelin whether GHSR was in DMoPC and DOPC nanodiscs. A slightly higher GTP turnover was nevertheless observed for GHSR in DMoPC, suggesting that the thickness of the membrane may affect, to some extent, ghrelin efficacy at activating Gi2.

The difference in G protein activation we observed could again result from differences either in G protein recruitment or in G protein activation. To analyze receptor:G protein interaction independently of G protein activation, we finally recorded the intermolecular GHSR:Gα LRET signal. The major component observed with GHSR in DMoPC nanodiscs, representing ca. 92% of the total population (Fig. 7e, Table 1), was the fast one that we previously assigned to the preassembled inactive GHSR:Gq complex[26]. This is consistent with the absence of any significant GTP turnover under such conditions. This suggests that the conformational state of the receptor achieved in DMoPC nanodiscs does not prevent preassembly of the receptor with Gq but abolishes the transition to an active complex. In contrast, in DOPC nanodiscs, a different LRET profile was obtained where the slow component corresponding to the active receptor:Gq complex was the major one (Fig. 7e, Table 1). A significant LRET signal was also obtained for Gi whether the ghrelin-loaded receptor was assembled in DMoPC or in DOPC nanodiscs, indicating that a receptor:G protein complex was formed in both cases (Fig. 7f). We previously demonstrated that Gi2 differs from

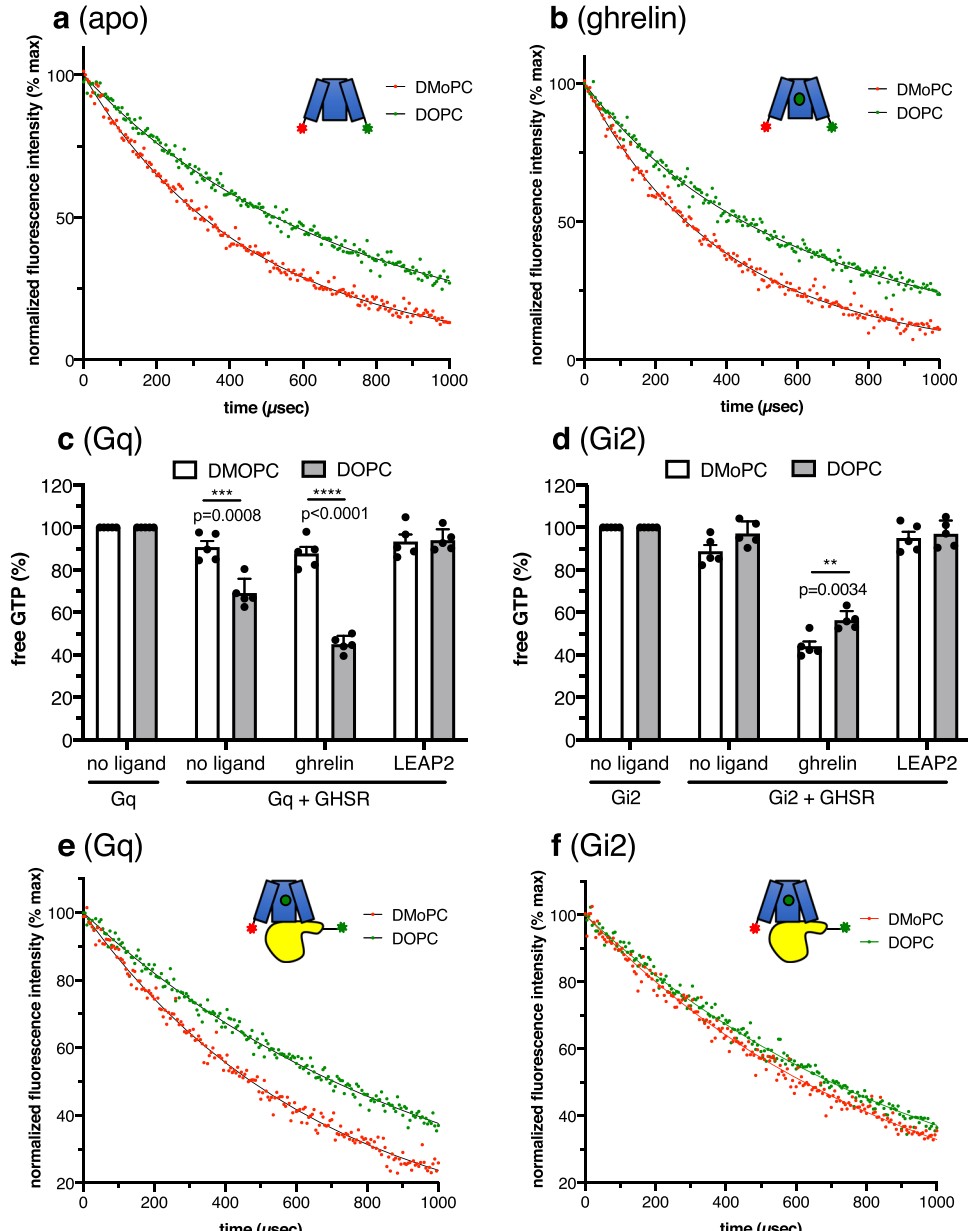

**Fig. 7 Membrane thickness affects GHSR conformational state and G protein selectivity.** Sensitized-emission decays from apo (**a**) or ghrelin-loaded (10 µM ligand) (**b**) GHSR labeled with the Lumi-4 Tb donor and Alexa Fluor 488 acceptor fluorophores in TM6 and TM1 and assembled into DMoPC or DOPC nanodiscs. The profiles were recorded in the absence of G protein. GTP turnover for Gq (**c**) and Gi2 (**d**) catalyzed by GHSR in DMoPC or DOPC nanodiscs in the absence of ligand, in the presence of 10 µM ghrelin or in the presence of 10 µM LEAP2(1-14). Luminescent signal was normalized to the signal obtained for the G protein in the absence of receptor. Data in (**c**) and (**d**) are mean ± SD of five experiments. Statistical values were obtained by means of unpaired Student's $t$ test (**$0.001 < p < 0.01$, ***$0.0001 < p < 0.001$, ****$p < 0.0001$). Intermolecular sensitized-emission decays from $G\alpha_q\beta_1\gamma_2$ (**e**) or $G\alpha_{i2}\beta_1\gamma_2$ (**f**) and ghrelin-loaded GHSR (10 µM ligand) assembled into DMoPC or DOPC nanodiscs, with the Lumi-4 Tb donor on the $G\alpha$ N-terminus and the Alexa Fluor 488 acceptor on the cytoplasmic end of TM1. Data in (**a**), (**b**), (**e**), and (**f**) are presented as normalized fluorescence intensity as a function of time and represent the average of three measurements. Source data are provided as a Source data file.

Gq in that it does not preassemble with isolated GHSR[26]. Besides, we observed a significant GTP turnover in both DMoPC and DOPC nanodiscs. Taken together, this indicates that the LRET signal in Fig. 7e is related to the formation of an active GHSR:Gi2 complex regardless of the nature of the lipid.

## Discussion
Using the nanodisc model membrane system, we found that PIP2 and GM3 selectively bind to GHSR. For PIP2, three different sites were identified that are all located in the cytoplasmic part of the

receptor, consistent with this lipid being essentially located in the inner leaflet of the plasma membrane of eukaryotic cells[38]. Although there are some differences that likely result from the diversity in protein sequence, most of the residues in the PIP2-binding sites of GHSR are homologous to those proposed to play a similar role in NTS1R and A2AR[9,17]. This suggests that PIP2 binding could be conserved through rhodopsin-like GPCRs. Upon binding, PIP2 reshapes the distribution of the different GHSR conformational states, favoring the active one with regard to G protein activation. Our data point to a functional mechanism based on a lipid-dependent shift of the inactive-to-active

equilibrium where the effect of PIP2 would occur because of the preferential binding of this lipid to the receptor active conformation, so that this conformation would attract the lipid, in turn favoring the corresponding conformational state. This could explain the increase in PIP2:GHSR FRET signal observed when cholesterol was included in the nanodiscs, as this sterol also appeared to stabilize the active state of GHSR (Supplementary Fig. 10). Of importance, this is very similar to the mechanism initially proposed for A2AR based on computational studies where PIP2 also preferentially bound to the receptor active state[17]. In the same way, differential recruitment of unsaturated lipids to the active and inactive states of A2AR has been proposed to affect the receptor activation state[32]. Besides PIP2, GHSR also binds GM3 at specific sites. GM3 was proposed to bind the adenosine and glucagon receptors and, by doing so, modulate their activation state[17,39]. Consistent with this computational data, GM3 was found here to bind at extracellular and intracellular sites of GHSR. The former sites were selective of GM3 while the latter were common with PIP2. The plasma membrane is asymmetric, with PIP2 and GM3 located exclusively in the internal and external leaflets, respectively. Although it cannot be excluded that GHSR displays some scramblase activity that would redistribute the lipids between the two leaflets[40], it is nevertheless likely that nanodiscs do not allow maintaining this asymmetry. Hence, a likely possibility would be that only the extracellular GM3 binding sites are physiologically relevant, as proposed for A2AR[17], and that binding to the intracellular region of GHSR only results from a cross-interaction with the PIP2-binding sites because of the lack of asymmetry in the nanodiscs. Of importance, these two classes of sites, i.e., the PIP2 intracellular sites and the GM3 extracellular ones, impact differentially on the conformation and functional properties of GHSR. Indeed, whereas PIP2 binding to the intracellular part of the receptor affects both the basal and the agonist-dependent activity, GM3 binding to the extracellular sites essentially affects agonist-dependent processes. Taken together, this illuminates a picture where GPCRs would exhibit an extended palette of specific allosteric binding sites for multiple lipid molecules. A concerted, dynamic, and specific modulation of that many protein:lipid interactions would increase target specificity and modulate the conformational landscapes of the receptors, ultimately adding a supplementary dimension to the way signaling is regulated.

Besides specific lipid:protein interactions, membrane thickness also affected GHSR pharmacology, in direct relationship with differences in the conformational landscape of the receptor. This is consistent with the recent structure of the NTS1R:Gi complex in nanodiscs where the lipid bilayer appeared to constrain the movements of TM6[41], highlighting the role of the bilayer in receptor conformational dynamics. In thin nanodisc membranes, the landscape of activated GHSR was populated with a major conformational state with an intermediate LRET value associated with a short distance between the cytoplasmic ends of TM1 and TM6, in the 35 Å range. To be noted, this is larger than the corresponding distance inferred from the crystal structure of inactive GHSR in its antagonist bound state (i.e., in the 28 Å range)[28], indicating that it may correspond to an intermediate state in the activation process (see below). Upon increasing the thickness of the bilayer, an alternative conformational state populated the active GHSR conformational landscape that was characterized by a low TM1-TM6 LRET efficiency, i.e., probably by an outward movement of the TM cytoplasmic domains of larger amplitude. As our measurements were carried out at the thermodynamic equilibrium, this could mean either that shortening the acyl chain of the lipids makes the "intermediate LRET" state more stable or that the hydrophobic mismatch in thin membranes increases the interconversion barrier between the

"intermediate-" and "low" LRET states. To be noted, only a few kcal/mol are sufficient to modify the interaction between transmembrane α-helices (in the 1–10 kcal/mol range[42]), which is in the order of what we found here. These changes in receptor conformation depending on the thickness of the membrane and their consequences on GHSR coupling selectivity raise some mechanistic questions. An intermediate FRET state with a GDP-loaded G protein had also been observed with the β2AR and Gs[43]. This state could be similar to the intermediate LRET state of GHSR coupled to GDP-loaded Gq, suggesting this could be possibly a quite general feature of the GPCR-catalyzed G protein activation process. It has been recently shown that GPCR coupling to G proteins may occur in a sequential manner with a series of intermediate receptor:G protein complexes that would be responsible for the selectivity in G protein coupling[44]. The conformational state we observe in DMoPC could thus correspond to such an intermediate state in the activation pathway of GHSR that would be more populated in DMoPC nanodiscs because of the particular features of the bilayer. If this is the case, then this state may be indeed associated with the selectivity process, as it is responsible for an efficient activation of Gi but for an inactive preassembly to Gq. This raises an additional point with regard to the differences in the conformational states of GHSR responsible for Gi and Gq activation. If the LRET efficiency is indeed related to the distance between the cytoplasmic tips of TM1 and TM6[26], as suggested by our MD simulations[26], then an outward movement of TM6 of lower amplitude would be sufficient for an efficient coupling of GHSR to Gi2 but not to Gq. This would be consistent with the different cryoEM structures where a small TM6 displacement is a hallmark of Gi coupling[45]. However, caution needs to be exerted as no such large difference was reported for M1R coupled to G11[46] or 5-HT2A in complex with a mini-Gαq/βγ heterotrimer[47]. Alternatively, the different movements in the TM domains could modulate the accessibility of the residues that are important for the selective coupling to the different G protein subtypes[48]. Interestingly, GTP turnover at Gi also seemed to slightly but significantly change depending on the membrane thickness (Supplementary Fig. 17), suggesting that, besides selectivity, efficacy at Gi activation could be somehow related, at least to some extent, to the amplitude of the TM6 outward movement, as previously proposed[49]. Whatever the details, a model emerges where the conformational dynamics of GPCRs would be accompanied by a variability in the structural arrangement of the G protein interaction domain in their cytoplasmic face, which in turn would mirror their selectivity of coupling to competing G proteins, in particular in the case of very promiscuous receptors such as GHSR. This would be in line with the mechanism recently proposed with the muscarinic receptor where the selective closure of the ligand-binding pocket trigger allosteric conformational changes in the cytoplasmic surface of the receptor that in turn control G protein coupling selectivity[50].

In closing, the levels and profiles of lipids are significantly altered by the nutritional intake in most of the physiopathological processes where ghrelin is directly involved. In addition, clinical work indicates that the responsiveness of GHSR to ghrelin is modified by alterations in circulating lipid profile[16]. Finally, several reports also demonstrated the importance of membrane composition for GHSR activation and desensitization[15]. It is thus tempting to speculate that GHSR-mediated signaling could in part depend on the dynamic regulation of the membrane composition, in response to different developmental and environmental conditions. This may apply to other GPCRs as, within a cell, all components are not uniformly mixed, but lipids and proteins form transient domains of specific composition and physicochemical properties, including bilayer thickness. In addition, the plasma membrane and intracellular compartments differ

in their lipid composition and in their physicochemical features such as thickness, and this may have an importance with regard to the increasing evidence for selective GPCR signaling in such compartments. Finally, cells are submitted to different stimuli, in particular mechanical stress, and at the molecular level, modulation in plasma membrane tension modulates its physicochemical features, in particular its local thickness[51]. All this points at lipids as being integral components of the signaling machinery, playing the role of spatially localized molecular switches that could control GPCR-dependent signaling.

## Methods

**Receptor preparation.** Human GHSR was expressed in *E. coli* inclusion bodies, folded in amphipol A8-35 from its SDS-unfolded state and then A8-35 was exchanged to β-DDM as described[22]. Evolved sortase (Addgene) and NW30 (DF/HCC plasmid depository) were expressed and purified as described[21]. Covalent circularization of NW30 was carried out as described[52]. Assembly into nanodiscs was carried out as follows. Solutions of lipids in chloroform (Avanti Polar Lipids) were mixed at the required composition (see text) and the solvent evaporated to form a lipid film. This film was solubilized by the addition of a 25 mM HEPES, 100 mM NaCl, 100 mM β-DDM buffer at a final 25 mM lipid concentration. Nanodisc assembly was carried out with the receptor immobilized on a solid matrix[7]. The His-tagged receptor in 25 mM HEPES, 100 mM NaCl, 2 mM β-DDM buffer was first bound onto a pre-equilibrated Ni-NTA superflow resin (Qiagen) at a protein-to-resin ratio at 0.1–0.2 mg of receptor per mL of slurry. The matrix-bound receptor was then mixed with 10 µM of the JMV3011 antagonist, the lipids, and cNW30 at a 0.1:1:1000 GHSR:cNW30:lipid molar ratio, and incubated under smooth stirring for 1 h at 4 °C. Polystyrene beads (Bio-Beads SM-2, BioRad) were added to the slurry. The amount of Bio-Beads was calculated based on the amount of detergent and of the β-DDM adsorptive capacity of the beads[53]. The mixture was incubated under smooth stirring for 4 h at 4 °C, the resin was extensively washed with a 50 mM Tris-HCl pH 8, 150 mM NaCl buffer and the His-tagged receptor finally eluted with the same buffer containing 200 mM imidazole. After extensive dialysis in 25 mM HEPES, 150 mM NaCl, 0.5 mM EDTA, pH 7.5, active receptor fractions were purified using affinity chromatography with the biotinylated version of JMV2959 immobilized on a streptavidin column[54]. The receptor in nanodiscs was loaded on the column, washed with a 25 mM HEPES, 150 mM NaCl, 0.5 mM EDTA, pH 7.5 buffer, and the proteins bound to the matrix recovered by washing the column with the same buffer containing 0.5 mM JMV2959. The latter was removed through extensive dialysis in 25 mM Na-HEPES, 150 mM NaCl, 0.5 mM EDTA, pH 7.5. We previously demonstrated using solution-state NMR and ligand-binding assays that the low-affinity antagonist JMV2959 was efficiently removed during this dialysis step[54]. GHSR-containing discs were separated from aggregates and possible trace amounts of ligand through a size-exclusion chromatography step on a Superdex 200 increase 10/300 GL column (GE Healthcare) using a 25 mM Na-HEPES, 150 mM NaCl, 0.5 mM EDTA, pH 7.5 buffer as the eluent. Receptor concentration was calculated from the know extinction coefficients of GHSR and cNW30, assuming a single receptor and two scaffolding proteins per nanodisc[22].

**Ligand-binding assays.** Competition binding assays were performed by measuring the fluorescence energy transfer signal between the purified receptor labeled with the lumi4-Tb NHS derivative at its N-terminus and a ghrelin peptide labeled with dy647 on an additional cysteine at its C-terminal[55] in the presence of increasing concentrations in competing compound[54]. Labeled ghrelin concentrations in the 0.1 µM range were used in all these assays. Binding curves were fit and analyzed with GraphPad Prism 8.

**Bodipy-FL PIP2 FRET binding assay.** Bodipy-FL PI (Echelon), Bodipy-FL PIP2 (Echelon), Bodipy-FL PA (Thermofisher), or C11 TopFluor GM3 (Thermofisher) were mixed with the unlabeled lipids at the desired molar ratio before the chloroform evaporation step. This lipid mixture was then used to assemble GHSR-containing nanodiscs as described above. Fluorescence was measured with a spectrometer with a pulsed Xe lamp as the excitation source ($\lambda_{exc}$: 337 nm) and an emission wavelength alternatively set at 490 and 520 nm. An Tr-FRET ratio was then obtained that corresponds to the ratio of the emitted intensities at these two wavelengths. The receptor concentration was set to 0.5 µM in all assays. The PIP2-to-GHSR and PA-to-GHSR molar ratios were calculated from the emission intensity of Bodipy-FL containing nanodiscs and a calibration curve obtained by measuring the emission intensity of solutions of Bodipy-FL-labeled lipids of known concentration in a buffer 25 mM HEPES, 100 mM NaCl, and 100 mM β-DDM (Supplementary Fig. 2). This was done on the assumption that the quantum yield of Bodipy-FL was the same whether the fluorophore was solubilized in β-DDM micelles or inserted into POPC bilayers.

**Coarse-grained molecular dynamics simulations.** CGMD simulations were performed using the MARTINI (v2.2) force field, using the ELNEDYN elastic network[56], and the GROMACS software (version 2020.3). The inactive state of

GHSR was retrieved from its available X-ray structure (PDB id 6KO5)[28] describing the receptor bound to a small antagonist and solved at 3.3 Å resolution. We then built an active model of GHSR by using the D2R:Gi complex as template (PDB id 6VMS)[57]. After alignment of the GHSR and D2R sequences, homology modeling was used to generate 100 models of GHSR with MODELLER v9.19[58]. The model displaying the best DOPE score was then selected as target conformation during a Targeted MD simulation (TMD). In a first step, the inactive conformation of GHSR was embedded in a lipid bilayer containing 156 POPC molecules, for a size of 80 Å in both *x* and *y* directions, then the system was neutralized and equilibrated with 0.15 M of NaCl using CHARMM-GUI[59]. All subsequent calculations were performed using the CHARMM36m force field[60]. The energy of the system was first minimized using 10,000 steps of conjugate gradient implemented in NAMD[61], before the full system was heated and equilibrated first in the NVT ensemble and second in the NPT ensemble. Here we followed the standard equilibration protocol suggested by CHARMM-GUI developers: 1876 ps at 1 atm. and 300 K. Finally, a TMD protocol was employed, applying a bias only on the GHSR residues Gly208 to Lys238, the target conformation being our homology model based on the active model of GHSR. All remaining atoms of GHSR were harmonically restrained in position using a force constant of 1 kcal.mol$^{-1}$.Å$^{-2}$, but residues Leu239 to Leu277 (ICL3), so that the loop could easily follow the motion of TM6. This protocol was designed to conserve at a maximum the atomic positions as described in the available X-ray structure of GHSR. The TMD simulation was performed in the NPT ensemble (1 atm., 300 K) over a period of 500 ps using an elastic constant of 200 kcal.mol$^{-1}$.Å$^{-2}$ scaled down by the number of selected atoms in TM6. The standard elastic network of MARTINI was applied to both the inactive and active conformations of the GHSR, thus preventing any large conformational changes. The main difference between these two conformations resides in the opening of TM5 + TM6. The resulting elastic networks allowed the "active" conformation to open or close freely whereas the "inactive" conformation was constrained in a close conformation due to the presence of elastics between TM6 and TM7. Each conformation of the receptor was embedded in a membrane model of size 100*100 Å$^2$ and composed of 7 PIP2 molecules on both leaflets, completed with POPC lipids using the CHARMM-GUI web-server[59]. For each inactive/active conformation, four other systems were also built (leading to a total of 10 different systems) by either replacing or mixing the 7 PIP2 molecules by/with POPG or GM3. Parameters of PIP2 (named POP2), and GM3 (named DPG3) were taken from the MARTINI force field[62]. Water and ions were finally added to complete the system and neutralize its global charge and to reach a concentration of 0.15 M of NaCl. After a short energy minimization, equilibration of each system was performed in the NVT ensemble for 4.75 ns, following the CHARMM-GUI protocol, i.e., decreasing progressively position restraints of both protein and membrane, and increasing the integration step from 2 to 20 fs. The production step was performed in the NPT ensemble using an integration step of 20 fs during 5 µs. Three replicas were run for each system, leading to 30 simulations for a total of 150 µs.

**GTP turnover assay.** GTP turnover was assessed as described[43,44,63,64]. All experiments were carried out at 15 °C. Briefly, the receptor was first incubated with the isolated G protein and, when applicable, the ligand (10 µM) for 30 min in a 25 mM HEPES, 100 mM NaCl, 5 mM MgCl$_2$, pH 7.5 buffer. We verified using receptor-free nanodiscs that Mg-triggered PIP2 clustering did not occur under such conditions, possibly because of the limited number of PIP2 molecules in the nanodiscs and to the fact that Mg$^{2+}$ has much weaker affinity and lower clustering propensity than other divalent cations such as Ca$^{[2+\ 38]}$. GTP turnover was then started by adding GTP (5 µM) and GDP (10 µM), and the amount of remaining GTP was assessed by measuring luminescence after 10 minutes incubation at 15 °C using the GTPase-Glo assay (Promega). The luminescence signal was normalized in each case to that in the absence of receptor (100%).

**GDP-release assay.** We developed an assay to qualitatively estimate whether GDP was released from the G protein upon complex formation with the receptor. To this end, Gα was loaded with MANT-GDP (ThermoFisher) during its last purification step by incubating the nucleotide-free protein with a 1.2-fold molar excess of MANT-GDP. Unbound nucleotide was eliminated by desalting on a ZebaSpin desalting spin column (ThermoFisher). The receptor in DMoPC or DOPC lipid nanodiscs was first incubated with 10 µM ghrelin and the G protein (1:1.5 receptor:G protein molar ratio) in a final 50 µL volume for 30 min at 15 °C. The mixture was then directly loaded on a S200 increase 5/150 column (GE Healthcare). The fluorescence emission of the eluted fractions was monitored at 448 nm with an excitation at 355 nm. The elution volume of free MANT-GDP was determined with the free nucleotide. The area under the free nucleotide peak was taken as a measure of the amount of released MANT-GDP. This value was normalized to 100% using as a reference the area of the free MANT-GDP peak obtained with the same amount of G protein in the absence of receptor after its unfolding with 6 M guanidinium chloride.

**Bimane labeling and fluorescence experiments.** Monobromobimane was introduced in a cysmin mutant of GHSR with a single reactive cysteine at position 255[6,27]. This mutant was shown to display native-like properties with regard to ligand-binding and G protein activation[26]. Labeling was carried out as previously

described[65]. Briefly, the receptor in cNW30 nanodiscs was incubated in the dark with a 1.5 molar excess in monobromobimane at 4 °C for 16 h. Unreacted dye was removed by extensive dialysis against a buffer 25 mM HEPES, 150 mM NaCl, 0.5 mM EDTA, pH 7.5.

**LRET measurements.** Labeling of Gα$_q$ and Gα$_{i2}$ on their N-terminus with Lumi-4-Tb cryptate (CisBio) was carried out using the NHS derivative of the fluorophore at neutral pH[26]. For labeling GHSR on Cys255[6.27], the purified receptor was incubated overnight at 16 °C in the presence of 100 μM TCEP. After desalting on a ZebaSpin column (ThermoFisher), 2 equivalents of Lumi-4-Tb maleimide were added for 10 min at room temperature and the labeled receptor desalted. The same procedure was applied to label C304[7.34] with Lumi-4-Tb maleimide. In this case, we used the cysminC304[7.34] mutant where the reactive cysteine C146[3.55] in ICL2 was substituted by a serine. We previously showed that, in this mutant, only C304[7.34] was labeled with the fluorophore[65]. Coupling of Alexa Fluor 488 to AzF was carried out by incubating the purified receptor in nanodiscs with Click-IT Alexa Fluor 488 DIBO Alkyne (ThermoFisher) at a final concentration of 300 μM at room temperature for 12 h. The AzF residue was initially introduced on GHSR in the cytoplasmic end of TM1 at position 71[1.60] using codon suppression technology[26]. The protein sample was desalted on a ZebaSpin column (ThermoFisher). Labeling ratios were calculated from the absorption spectra of the labeled proteins using the known extinction coefficients of the receptor and the fluorophores. LRET was measured with a spectrometer with a pulsed Xe lamp as the excitation source ($\lambda_{exc}$: 337 nm). To avoid any artifact related to an incomplete labeling, the donor lifetimes in the presence of the acceptor were measured through the acceptor-sensitized emission at 515 nm. Sensitized emission was fitted to a two exponential decay function, and the goodness of the exponential fit was determined from the random residual. The slow and fast components correspond to the two time constants of donor fluorescence decay inferred from these exponentials. Molecular fractions were calculated from the pre-exponential factors and the excited-state lifetime values[66] and the distances between the donor and acceptor fluorophores estimated from the LRET lifetime ($\tau_{ad}$) and donor-only lifetime ($\tau_d$) using the Förster equation, as described[26].

**Statistics.** Statistical analyses were all carried out with GraphPad Prism 8.

**Reporting summary.** Further information on research design is available in the Nature Research Reporting Summary linked to this article.

## Data availability
Data supporting the findings of this manuscript are available from the corresponding author upon reasonable request. A reporting summary for this Article is available as a Supplementary Information file. The PDB files that were analyzed have been published before and were obtained from the RCSB Protein Data Bank using the accession codes PDB 6KO5 and 6VMS.

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

## Acknowledgements

This work was supported by CNRS, Université de Montpellier and the Agence Nationale de la Recherche (ANR-17-CE11-0011, ANR-17-CE18-0022). We thank CAMPUS FRANCE for promoting the French-Brazilian collaboration via the CAPES-COFECUB project number Ph-C882/17. We also thank GENCI (Grand Équipement National de Calcul Intensif) and TGCC/IDRIS to have selected us for the "Great Challenge" phases of both the IRENE JOLIOT-CURIE and JEAN-ZAY supercomputers.

## Author contributions

J.L.B., J.A.F., M.D., N.F., M.L. and L.J.C. designed the research. M.D., C.M.K., M.L. and A.A.S.G. performed the research. S.D., S.C. and J.A.F. provided materials and reagents. M.D., M.L., A.A.S.G., C.M.K., S.D., S.C., S.M., P.M.B., J.A.F., L.J.C., N.F. and J.L.B. analyzed the data. J.L.B. wrote the paper with input of all other authors.

## Competing interests

The authors declare no competing interests.
