## [Peer Review File · Nature Communications]

Reviewer #1 (Remarks to the Author):

This manuscript discusses how lipids influence the conformation, G protein selectivity and activity of GPCRs by using artificial nanodiscs and FRET systems. The target GPCR by the authors is the ghrelin receptor, which couples mainly with Gq to regulate many physiological functions such as GH release, appetite stimulation, and glucose metabolism, etc.

The authors found that PIP2 binds to the ghrelin receptor at the specific sites and changes the conformation into an active/active-like form. In addition, they found that membrane thickness influences the conformational repertoire of the receptor and changes G protein selectivity.

The results are interesting, because this is the first report about the interaction and conformational changes by lipids on Gq coupled class A GPCRs. However, some of the results are partially overlapped with the other recent papers, which weakened the novelty of this manuscript.

Major points

1, Several papers have already revealed the interactions between GPCRs and lipids, and examined functions of GPCRs, though the methods used in these papers are different (NMR and MD simulation). For example, Yen et al. (reference #7) reported that PIP2 stabilizes active states of GPCRs (A2AR and NTSR1) and enhances selectivity of G proteins. The reference #9 and #22 also discussed about lipid interactions with GPCRs.

What are the difference or similar points between the present paper and the some referenced papers? In particular, the differences about the binding sites of PIP2 and the G protein coupling selections should be discussed.

2, Acyl-modification of ghrelin (n-octanoic acid) is essential for ghrelin's activity. Then, is there any differences in a lipid binding mode and an active state of the ghrelin receptor by binding of the acyl-modified ghrelin or MK0677 (no acyl-modification)?

3, It is reported that the ghrelin receptor not only couples with Gq and Gi2/o but also with G12/13. Does PIP2 modulate G12/13 activation? What is the changes of GTP turnover with G12/13 activation?

4, Lane 160-161 and figure 2f. The authors described as "Mutating sites 1 or 2 led to a ca. 50% decrease in the signal, while the mutating S3 was associated with a lower decrease in the FRET signal, in the 10% range". However, from the figure 2f, it seems to me that the decreases in the signal are 50% for S1, 40% for S2 and 20% for S3, respectively.

5, In figure 3a, the authors described as "whereas binding of the inverse agonist LEAP2(1-14) resulted in an opposite change" (lane 177). However, from the figure 3a, it is hard to find the differences of the emission intensity between LEAP2 and LEAP2-PIP2.

6, In the figures of the emission decay (figs. 3d, 5e, 5f, 6e, 6f), how does the author distinguish between the slow and fast components? Please explain about the decay time components.

7, In the lane 314, the authors described as "modifying the membrane thickness did not significantly alter Gi2 activation". Do the authors have any interpretation about the different activations between Gq and Gi2/o by the membrane thickness?

8, In figure 4b, it seems to me that there are statistical significance differences between the values

of λ_{\max} in the presence of PIP2 and no PIP2 in wild type and A204E by ghrelin. In addition, in figures 5a and 5b, are there any statistical significance differences between no PIP2 and the presence of PIP2 in (+GHSR+ghrelin)? Also in the figure 6c and 6d, there are no statistical significance differences. Methods for statistical analysis should be added.

9, Lane 527. PDB ID of NTSR1 should be added. The authors probably construct the GHSR model based on an inactive form of NTSR1.

Minor points

1, Lane 333, Figure 6e? I think this is Figure 6d.

2, Lane 478. Please cut “in” from “described in”.

3, Lane 665, Figure legend of figure 4. “of” should be “or”.

4, In the supplementary figures and legends, the alphabets should be changed to the small letters. In the text, the alphabets are described as the small letters. For example lane 80, there is “Fig. S1a” and lane 81 “Fig.S1b,c”.

5, In the figure S3, an oxygen is missing in the acyl chain of PI.

6, Lane 91. MS should be described as MS (Mass Spectrometry).

7, Lane 140, “(superscript numbers follow Ballesteros-Weinstein numbering)” should be move to the lane 93 after “cysteine mutant”.

8, Lane 139. the BW number for R72 is missing.

9, In the text, the authors should unify the ghrelin receptor to GHSR.

Reviewer #2 (Remarks to the Author):

Review of Nature Communications manuscript NCOMMS-20-42334-T

This is an interesting manuscript on an important topic, namely how lipids regulate the activity of GPCRs. The specific system studied is the interactions of PIP2 with the ghrelin receptor (GHSR). A combination of experimental (FRET of GHSR in nanodiscs) and computational (coarse grained MD) approaches is employed. It is shown that PIP2 allosterically favours activation of the receptor by preferential binding to cytoplasmic sites in the active state protein. Membrane thickness also seems to play a role in terms of G protein selectivity. This study is also of potential biomedical interest given the role of GHSR in metabolic regulation.

Major comments.

1. Overall, these studies provide a persuasive account of the allosteric regulation of a GPCR by lipids

and/or its membrane environment. My main concern is that the simulation studies are performed with a homology model of the GHSR rather than the recent structure. This may not be too much of a problem given the simulations are at CG resolution, but I think they should run additional simulations with the new GHSR structure to demonstrate this is indeed the case.

2. The receptor is immobilized on a solid matrix. Can we be certain that this will not perturb the conformational equilibria of the receptor, and the sensitivity of this to lipids?

3. Page 4: the FRET signal reaches a maximum at PIP2:GHSR of about 4 or 5. How does this compare with the mean number of PIP2 molecules bound per receptor in the simulations? (on page 5 it is stated this ratio is “fully consistent” with 5 or 6 PIP2 molecule bound in the simulations)

4. In the FRET experiments neither POPs or POPG competes with PIP2. Does either PS or PG binding to the receptor occur in the simulations?

5. The major conclusion from the FRET studies is that PIP2 binds preferentially to the active state of the GHSR (page 6). This is an important result. Interestingly it agrees well with the computational results in Ref 22 (for the A2AR) which predicted such a difference in PIP2 binding to active and inactive states, and so this should be mentioned. Also, it should be possible to build a homology model of active state GHSR (especially as their simulations are based on a homology model of the inactive state) and show by simulations that PIP2 binds more strongly to the active state model, thus reinforcing this interpretation of their FRET studies.

6. The author tried to find a correlation between the membrane thickness and receptor activation efficiency by using only two lipid species of different lengths. Is it possible to explore an additional lipid with a tail length other than 14C to 18C to make this correlation more persuasive?

7. (Line 279 P.8) The authors suggest that DMOPC (14C) and DOPC (18C) should have the same membrane fluidity because they display the same unsaturation degree. However, lipid tail length plays a major part in determining the membrane fluidity as well.

Minor comments:

Some work is needed to make Fig. 2 a bit clearer, especially to the non-specialist reader. Specifically:

- Fig 2a/b – It is unclear what this density is supposed to show. It looks like the authors need to change the isovalue of the density in the VMD settings. This also looks like the ‘density’ has been calculated on a single simulation snapshot rather than by averaging over the trajectory but this might just be a problem with the selected isovalue.

- Fig 2c-e – This might be clearer if the authors show the PIP2 using the `cg_bonds` script in MARTINI rather than as unconnected VDW spheres. Also, specify the colours used in the figure legend.

Reviewer #3 (Remarks to the Author):

The manuscript by Damian et al provides a detailed description of how the structure and function of GSHR are affected by lipidic components.

They conclude that PIP2 bind three specific sites on the cytoplasmic side of the receptor and that its presence of PIP2 favors the active state of the receptor. They also observe that shortening the bilayer thickness affects G protein coupling. They conclude that the functional profile of the receptor may vary with cell types.

The experiments are sound, well-crafted and overall support the conclusions.

However, in my opinion, some points need clarification or improvement.

1/ The first issue is the scope/structure of the paper. The title "Allosteric modulation of ghrelin receptor 1 signaling by lipids" is *stricto sensu* correct (PIP2 is a lipid indeed) but seems to suggest that the authors will investigate the effect of various lipids while they focus on PIP2. Considering that, along with PIP2, other lipids can modulate GPCRs, it is not quite clear while they aren't included. The specific displacement of the labeled PIP2 by unlabelled PIP2 but not by other unlabelled lipids indicate that PIP2 indeed interacts but it does not mean that other lipids do or do not.

In terms of structure, I find somewhat unclear why, after focusing on PIP2, the authors then look at changing the bilayer thickness. These seems like somewhat separated studies glued together.

2/ Can the authors provide some information regarding the labelled analogs.

I have been struggling in looking for the exact structure of the labeled lipids on the thermofisher website, but I assume they in fact come from Echelon where BODIPY-FL PIP2 contains two peptide bonds. From what I could find the BODIPY-FL PA does not. This could affect the integration in the bilayer, the location of the probe and therefore the FRET signal.

Could that affect the quantum yield of the probes (I guess that Figure S2 suggests it doesn't)?

3/ On Figure 1a, for lipid:protein ratio above 5 it appears that this fluorescence value goes up with "a slope similar to that of the control lipid Bodipy-FL PA, suggesting a random distribution of the additional fluorescent PIP2 molecules in the nanodisc". This seems odd to me. Shouldn't we assume that if all three binding sites are occupied maximum FRET efficacy is achieved or that, at least, distantly located FRET acceptors should not contribute ?

4/ The choice of Lumi4-Tb as FRET donor seems odd to me. Sure, the extended lifetime allows to decrease background, but the probe is very large and, being directly coupled to the bottom of TM6 one could fear that it may affect the dynamics of the receptor. The authors should at least comment on that.

Regarding LRET, the authors also rely on the reader's preliminary knowledge of their previous (2015?) papers to know what LRET means/is and permit to reveal. A bit of more explanation on the information provided by the sensitized-emission decay would be useful as it is far from intuitive.

5/ The authors conclude that PIP2 favors the active state and show that, conversely, "PIP2

preferentially binds to the active state of GHSR” but showing lack of interaction on a mutant devoid of constitutive activity. Wouldn't this better demonstrated by monitoring GSHR:PIP2 FRET signal in the absence and presence of the inverse agonist LEAP2? This appears to be missing from Figure 4.

6/ The effect of PIP2 on G coupling is interesting but somewhat weak (especially in light of the MB data). It seems that the differences in GTP turnover are not statistically significant. If so, the authors should comment.

7/ Regarding the cytoplasmic vs extracellular location of the binding sites, Fig S4 lacks important controls/information. The methods do not specify how lumi4-Tb was attached to either Cys255 or Cys304 but I assume by a thiol-reactive group such as maleimide. The labelling details are not provided either.

In my experience that reactivity of target cysteine to labels vary greatly, sometimes as much as 1 in 10. I also note that Cys304 appears to be located in/at the edge of the bilayer and thus poorly accessible. The authors much show labelling and fluorescence of C304 and not just lack of transfer.

8/ Model vs. crystal

I am surprised that the authors did not correct their modelling by using the recently published structure of GSHR (though I understand that it requires extra work). There are some differences between NTS1 and GSHR at the proposed PIP2 binding sites that would justify re-running the coarse-grained simulations. On the other hand, the authors may argue that the model was “good enough” to identify experimentally-validated PIP2 binding sites, but then again they could have tested the same sites with a literature survey (which they did perform).

9/ The physiological relevance of the study on the effect of bilayer thickness remains unclear to me. To my knowledge it is not expected that bilayer thickness will change significantly from one cell type to another, whether through acyl chain length distribution or temperature changes. Furthermore, the fact that helix-helix distances were found to be shorter in DMOPC than in DOPC seems at odds with studies on other membrane proteins that indicate increase in tilt angle with decreased thickness. While it is possible that the decreased TM1-TM6 distance is due to a tilt change the overall interpretation remains vague. Nevertheless the fact that the 2 G protein are affected differently is intriguing, indeed possibly linked to pre-assembly.

10/ The Discussion should be shortened. The first parts is just a repeat of the Result section. The last paragraphs are generalities on membrane variations. I don't think that Figure 7 brings much.

Overall, I think this is a nice study but the focus should be clearer. I personally would have liked to see other lipids being tested as I'm fearing that importance PIP2 is being over represented in the field.

REVIEWER COMMENTS

Reviewer #1 (Remarks to the Author):

This manuscript discusses how lipids influence the conformation, G protein selectivity and activity of GPCRs by using artificial nanodiscs and FRET systems. The target GPCR by the authors is the ghrelin receptor, which couples mainly with Gq to regulate many physiological functions such as GH release, appetite stimulation, and glucose metabolism, etc.

The authors found that PIP2 binds to the ghrelin receptor at the specific sites and changes the conformation into an active/active-like form. In addition, they found that membrane thickness influences the conformational repertoire of the receptor and changes G protein selectivity.

The results are interesting, because this is the first report about the interaction and conformational changes by lipids on Gq coupled class A GPCRs. However, some of the results are partially overlapped with the other recent papers, which weakened the novelty of this manuscript.

Major points

1, Several papers have already revealed the interactions between GPCRs and lipids, and examined functions of GPCRs, though the methods used in these papers are different (NMR and MD simulation). For example, Yen et al. (reference #7) reported that PIP2 stabilizes active states of GPCRs (A2AR and NTSR1) and enhances selectivity of G proteins. The reference #9 and #22 also discussed about lipid interactions with GPCRs. What are the difference or similar points between the present paper and the some referenced papers? In particular, the differences about the binding sites of PIP2 and the G protein coupling selections should be discussed.

2, Acyl-modification of ghrelin (n-octanoic acid) is essential for ghrelin's activity. Then, is there any differences in a lipid binding mode and an active state of the ghrelin receptor by binding of the acyl-modified ghrelin or MK0677 (no acyl-modification)?

3, It is reported that the ghrelin receptor not only couples with Gq and Gi2/o but also with G12/13. Does PIP2 modulate G12/13 activation? What is the changes of GTP turnover with G12/13 activation?

4, Lane 160-161 and figure 2f. The authors described as "Mutating sites 1 or 2 led to a ca. 50% decrease in the signal, while the mutating S3 was associated with a lower decrease in the FRET signal, in the 10% range". However, from the figure 2f, it seems to me that the decreases in the signal are 50% for S1, 40% for S2 and 20% for S3, respectively.

5, In figure 3a, the authors described as "whereas binding of the inverse agonist LEAP2(1-14) resulted in an opposite change" (lane 177). However, from the figure 3a, it is hard to find the differences of the emission intensity between LEAP2 and LEAP2-PIP2.

6, In the figures of the emission decay (figs. 3d, 5e, 5f, 6e, 6f), how does the author distinguish between the slow and fast components? Please explain about the decay time components.

7, In the lane 314, the authors described as "modifying the membrane thickness did not significantly alter Gi2 activation". Do the authors have any interpretation about the different activations between Gq and Gi2/o by the membrane thickness?

8, In figure 4b, it seems to me that there are statistical significance differences between

the values of λ_{max} in the presence of PIP2 and no PIP2 in wild type and A204E by ghrelin. In addition, in figures 5a and 5b, are there any statistical significance differences between no PIP2 and the presence of PIP2 in (+GHSR+ghrelin)? Also in the figure 6c and 6d, there are no statistical significance differences.

Methods for statistical analysis should be added.

9, Lane 527. PDB ID of NTSR1 should be added. The authors probably construct the GHSR model based on an inactive form of NTSR1.

Minor points

1, Lane 333, Figure 6e? I think this is Figure 6d.

2, Lane 478. Please cut "in" from "described in".

3, Lane 665, Figure legend of figure 4. "of" should be "or".

4, In the supplementary figures and legends, the alphabets should be changed to the small letters. In the text, the alphabets are described as the small letters. For example lane 80, there is "Fig. S1a" and lane 81 "Fig.S1b,c".

5, In the figure S3, an oxygen is missing in the acyl chain of PI.

6, Lane 91. MS should be described as MS (Mass Spectrometry).

7, Lane 140, "(superscript numbers follow Ballesteros-Weinstein numbering)" should be move to the lane 93 after "cysteine mutant".

8, Lane 139. the BW number for R72 is missing.

9, In the text, the authors should unify the ghrelin receptor to GHSR.

Reviewer #2 (Remarks to the Author):

Review of Nature Communications manuscript NCOMMS-20-42334-T

This is an interesting manuscript on an important topic, namely how lipids regulate the activity of GPCRs. The specific system studied is the interactions of PIP2 with the ghrelin receptor (GHSR). A combination of experimental (FRET of GHSR in nanodiscs) and computational (coarse grained MD) approaches is employed. It is shown that PIP2 allosterically favours activation of the receptor by preferential binding to cytoplasmic sites in the active state protein. Membrane thickness also seems to play a role in terms of G protein selectivity. This study is also of potential biomedical interest given the role of GHSR in metabolic regulation.

Major comments.

1. Overall, these studies provide a persuasive account of the allosteric regulation of a GPCR by lipids and/or its membrane environment. My main concern is that the simulation studies are performed with a homology model of the GHSR rather than the recent structure. This may not be too much of a problem given the simulations are at CG resolution, but I think they should run additional simulations with the new GHSR structure to demonstrate this is indeed the case.

2. The receptor is immobilized on a solid matrix. Can we be certain that this will not perturb the conformational equilibria of the receptor, and the sensitivity of this to lipids?

3. Page 4: the FRET signal reaches a maximum at PIP2:GHSR of about 4 or 5. How does this compare with the mean number of PIP2 molecules bound per receptor in the simulations? (on page 5 it is stated this ratio is “fully consistent” with 5 or 6 PIP2 molecule bound in the simulations)

4. In the FRET experiments neither POPs or POPG competes with PIP2. Does either PS or PG binding to the receptor occur in the simulations?

5. The major conclusion from the FRET studies is that PIP2 binds preferentially to the active state of the GHSR (page 6). This is an important result. Interestingly it agrees well with the computational results in Ref 22 (for the A2AR) which predicted such a difference in PIP2 binding to active and inactive states, and so this should be mentioned. Also, it should be possible to build a homology model of active state GHSR (especially as their simulations are based on a homology model of the inactive state) and show by simulations that PIP2 binds more strongly to the active state model, thus reinforcing this interpretation of their FRET studies.

6. The author tried to find a correlation between the membrane thickness and receptor activation efficiency by using only two lipid species of different lengths. Is it possible to explore an additional lipid with a tail length other than 14C to 18C to make this correlation more persuasive?

7. (Line 279 P.8) The authors suggest that DMOPC (14C) and DOPC (18C) should have the same membrane fluidity because they display the same unsaturation degree. However, lipid tail length plays a major part in determining the membrane fluidity as well.

Minor comments:

Some work is needed to make Fig. 2 a bit clearer, especially to the non-specialist reader. Specifically:

- Fig 2a/b – It is unclear what this density is supposed to show. It looks like the authors need to change the isovalue of the density in the VMD settings. This also looks like the ‘density’ has been calculated on a single simulation snapshot rather than by averaging over the trajectory but this might just be a problem with the selected isovalue.
- Fig 2c-e – This might be clearer if the authors show the PIP2 using the cg_bonds script in MARTINI rather than as unconnected VDW spheres. Also, specify the colours used in the figure legend.

Reviewer #3 (Remarks to the Author):

The manuscript by Damian et al provides a detailed description of how the structure and function of GSHR are affected by lipidic components.

They conclude that PIP2 bind three specific sites on the cytoplasmic side of the receptor and that its presence of PIP2 favors the active state of the receptor. They also observe that shortening the bilayer thickness affects G protein coupling. They conclude that the functional profile of the receptor may vary with cell types.

The experiments are sound, well-crafted and overall support the conclusions.

However, in my opinion, some points need clarification or improvement.

1/ The first issue is the scope/structure of the paper. The title “Allosteric modulation of

ghrelin receptor 1 signaling by lipids" is *stricto sensu* correct (PIP2 is a lipid indeed) but seems to suggest that the authors will investigate the effect of various lipids while they focus on PIP2. Considering that, along with PIP2, other lipids can modulate GPCRs, it is not quite clear while they aren't included. The specific displacement of the labeled PIP2 by unlabelled PIP2 but not by other unlabelled lipids indicate that PIP2 indeed interacts but it does not mean that other lipids do or do not.

In terms of structure, I find somewhat unclear why, after focusing on PIP2, the authors then look at changing the bilayer thickness. These seems like somewhat separated studies glued together.

2/ Can the authors provide some information regarding the labelled analogs.

I have been struggling in looking for the exact structure of the labeled lipids on the thermofisher website, but I assume they in fact come from Echelon where BODIPY-FL PIP2 contains two peptide bonds. From what I could find the BODIPY-FL PA does not. This could affect the integration in the bilayer, the location of the probe and therefore the FRET signal.

Could that affect the quantum yield of the probes (I guess that Figure S2 suggests it doesn't)?

3/ On Figure 1a, for lipid:protein ratio above 5 it appears that this fluorescence value goes up with "a slope similar to that of the control lipid Bodipy-FL PA, suggesting a random distribution of the additional fluorescent PIP2 molecules in the nanodisc". This seems odd to me. Shouldn't we assume that if all three binding sites are occupied maximum FRET efficacy is achieved or that, at least, distantly located FRET acceptors should not contribute ?

4/ The choice of Lumi4-Tb as FRET donor seems odd to me. Sure, the extended lifetime allows to decrease background, but the probe is very large and, being directly coupled to the bottom of TM6 one could fear that it may affect the dynamics of the receptor. The authors should at least comment on that.

Regarding LRET, the authors also rely on the reader's preliminary knowledge of their previous (2015?) papers to know what LRET means/is and permit to reveal. A bit of more explanation on the information provided by the sensitized-emission decay would be useful as it is far from intuitive.

5/ The authors conclude that PIP2 favors the active state and show that, conversely, "PIP2 preferentially binds to the active state of GHSR" but showing lack of interaction on a mutant devoid of constitutive activity. Wouldn't this better demonstrated by monitoring GSHR:PIP2 FRET signal in the absence and presence of the inverse agonist LEAP2? This appears to be missing from Figure 4.

6/ The effect of PIP2 on G coupling is interesting but somewhat weak (especially in light of the MB data). It seems that the differences in GTP turnover are not statistically significant. If so, the authors should comment.

7/ Regarding the cytoplasmic vs extracellular location of the binding sites, Fig S4 lacks important controls/information. The methods do not specify how lumi4-Tb was attached to either Cys255 or Cys304 but I assume by a thiol-reactive group such as maleimide. The labelling details are not provided either.

In my experience that reactivity of target cysteine to labels vary greatly, sometimes as much as 1 in 10. I also note that Cys304 appears to be located in/at the edge of the bilayer and thus poorly accessible. The authors much show labelling and fluorescence of

C304 and not just lack of transfer.

8/ Model vs. crystal

I am surprised that the authors did not correct their modelling by using the recently published structure of GSHR (though I understand that it requires extra work). There are some differences between NTS1 and GSHR at the proposed PIP2 binding sites that would justify re-running the coarse-grained simulations. On the other hand, the authors may argue that the model was "good enough" to identify experimentally-validated PIP2 binding sites, but then again they could have tested the same sites with a literature survey (which they did perform).

9/ The physiological relevance of the study on the effect of bilayer thickness remains unclear to me. To my knowledge it is not expected that bilayer thickness will change significantly from one cell type to another, whether through acyl chain length distribution or temperature changes. Furthermore, the fact that helix-helix distances were found to be shorter in DMOPC than in DOPC seems at odds with studies on other membrane proteins that indicate increase in tilt angle with decreased thickness. While it is possible that the decreased TM1-TM6 distance is due to a tilt change the overall interpretation remains vague. Nevertheless the fact that the 2 G protein are affected differently is intriguing, indeed possibly linked to pre-assembly.

10/ The Discussion should be shortened. The first parts is just a repeat of the Result section. The last paragraphs are generalities on membrane variations. I don't think that Figure 7 brings much.

Overall, I think this is a nice study but the focus should be clearer. I personally would have liked to see other lipids being tested as I'm fearing that importance PIP2 is being over represented in the field.

POINT-BY-POINT ANSWER TO REVIEWER COMMENTS

Before anything else, we would like to thank the three expert reviewers for taking the time to read the manuscript, and for providing helpful and insightful comments that helped us significantly increase the quality of the work. In particular, the proposal to extend the scope of the paper to other compounds besides PIP2 led us to investigate the effects of another lipid, GM3. Our manuscript thus now includes a new series of experiments with this lipid that indicate the ghrelin receptor displays a palette of specific sites for distinct lipids that may differentially regulate the functional and structural properties of this GPCR. Besides, as the statistical significance of some of our data was an appropriate concern for all the reviewers, we carried out additional measurements to get better statistics, especially for the receptor-catalyzed GTP turnover assays. Finally, we repeated all the simulations with the crystal structure of the ghrelin receptor instead of the neurotensin receptor-based homology model. Below is detailed how we addressed all the points the reviewers raised. These include specific comments and additional experiments.

Reviewer #1:

Major comments

1, Several papers have already revealed the interactions between GPCRs and lipids, and examined functions of GPCRs, though the methods used in these papers are different (NMR and MD simulation). For example, Yen et al. (reference #7) reported that PIP2 stabilizes active states of GPCRs (A2AR and NTSR1) and enhances selectivity of G proteins. The reference #9 and #22 also discussed about lipid interactions with GPCRs. What are the differences or similar points between the present paper and the some referenced papers? In particular, the differences about the binding sites of PIP2 and the G protein coupling selections should be discussed.

Answer: as far as the PIP2 binding sites are considered, the general architecture of the sites reported for NTS1R and A2AR in the papers the reviewer refers to are closely related to those we identified here for GHSR. There are subtle differences in the residues involved, however. In particular, the “second” site in NTS1R is essentially composed of residues from TM4 (reference # 7 in our manuscript) whereas, in GHSR, it is formed by residues from TM5 and TM6. Similar residues in TM5 and TM6 were nevertheless proposed to be involved in PIP2 binding to A2AR, based on computational studies (reference #22). Taken together, although there may be subtle differences in the residues involved in PIP2 binding because of the particular sequences of the receptors, all the data nevertheless converges toward specific PIP2:GPCR interactions in the intracellular regions of the receptors, *i.e.*, with PIP2 molecules located in the lower leaflet of the lipid bilayer. This is also the case for the new data with GM3 we included in the revised version of the manuscript where similar extracellular binding sites were proposed for A2AR, based on computational analyses (reference #22). With regard to the functional consequences of the interaction with PIP2, there are also subtle differences between our observations and those reported in reference #7. Indeed, in the latter, PIP2 was proposed to increase the stability of the receptor:G protein complex whereas our data point to an impact of PIP2 on the stability of the active state of the receptor, independently of the G protein. But our observations are again totally in line with the computational studies with A2AR that also indicated a privileged interaction of PIP2 with the active receptor conformation (see reviewer #2 comments). In any case, these differences converge to a similar functional output, *i.e.*, an increase in G protein activation. As proposed by the reviewer, this has been added to the discussion section of the revised version of our manuscript.

2, *Acyl-modification of ghrelin (n-octanoic acid) is essential for ghrelin's activity. Then, is there any difference in a lipid binding mode and an active state of the ghrelin receptor by binding of the acyl-modified ghrelin or MK0677 (no acyl-modification)?*

Answer: this a very good remark, as a possible interaction of the lipid moiety of ghrelin with the nanodiscs could have perturbed the arrangement of the bilayer and therefore indirectly affected GHSR functioning. As proposed by the reviewer, we carried out the monobromobimane (MB) emission experiments and GTP turnover assays in the absence and presence of PIP2 with a non-peptide GHSR agonist, MK0677, instead of acylated ghrelin. We also included in these assays another non-peptide full agonist of GHSR developed in our laboratory, JMV1843. The effects of PIP2 were the same whether acylated ghrelin, MK0677 or JMV1843 were considered, suggesting that the lipid chain of ghrelin has no major impact on these effects. This data has been added to the revised version of the manuscript (**Supplementary Fig.9**).

3, *It is reported that the ghrelin receptor not only couples with Gq and Gi2/o but also with G12/13. Does PIP2 modulate G12/13 activation? What is the changes of GTP turnover with G12/13 activation?*

Answer: Unfortunately, we were not able to carry out the proposed experiments, as we could not produce G13 as a purified protein. Indeed, this G protein subtype is particularly difficult to express as a recombinant protein in sufficient amounts, whatever the expression system. A commercial source was available from Kerafast that we used in our previous work with GHSR (M'Kadmi et al. (2015). *J Biol Chem* **290**, 27021). However, this protein doesn't seem to be commercialized anymore, so that we could not carry out the experiments proposed. We apologize for that. A sentence has been added to the results section to take into account this remark ("*GHSR also couples to G13 but coupling to this G protein subtype could not be analyzed here, as we could not get purified G13 for the GTP turnover assays.*").

4, *Lane 160-161 and figure 2f. The authors described as "Mutating sites 1 or 2 led to a ca. 50% decrease in the signal, while the mutating S3 was associated with a lower decrease in the FRET signal, in the 10% range". However, from the figure 2f, it seems to me that the decreases in the signal are 50% for S1, 40% for S2 and 20% for S3, respectively.*

Answer: this was indeed a rough approximation of the data in **Figure 2f** (**Fig.2d** in the revised manuscript). We modified the values accordingly in the revised version of our manuscript.

5, *In figure 3a, the authors described as "whereas binding of the inverse agonist LEAP2(1-14) resulted in an opposite change" (lane 177). However, from the figure 3a, it is hard to find the differences of the emission intensity between LEAP2 and LEAP2-PIP2.*

Answer: Our sentence was confusing and we apologize for that. In fact, the sentence "*whereas binding of the inverse agonist LEAP2(1-14) resulted in an opposite change*" referred to the intrinsic effect of LEAP2 on GHSR conformation, *i.e.*, the difference between the apo- and LEAP2-loaded states, independently of PIP2. In this case, indeed, there is a significant difference between the MB emission spectra in the absence of ligand and in the presence of LEAP2(1-14). In contrast, as perfectly noted by the reviewer, there is no significant difference in the emission intensity between

LEAP2 and LEAP2-PIP2, which is consistent with our model suggesting a preferential interaction of PIP2 with the active state of the receptor. We modified the sentence accordingly so that we hope there is no more ambiguity (“...whereas binding of the inverse agonist LEAP2(1-14) resulted in an opposite change when compared to the apo receptor (Fig.3a,b).”). In addition, we tried to make the following sentence clearer (“Incorporation of 2.5% PIP2 into the lipid nanodiscs resulted in an additional decrease in the emission intensity and increase in λ_{max} for the apo and ghrelin-loaded GHSR. In contrast, no significant effect was observed for LEAP2 loaded receptor (Fig.3a,b), suggesting that the effects of PIP2 were somehow related to the active state of GHSR (see below”).

6, In the figures of the emission decay (figs. 3d, 5e, 5f, 6e, 6f), how does the author distinguish between the slow and fast components? Please explain about the decay time components.

Answer: the fluorescence decays were fitted to a sum of two discrete exponentials. The slow and fast components then correspond to the two time constants of the donor fluorescence decay inferred from these exponentials. A sentence has been added to the revised version of the manuscript (Methods section), with the corresponding reference.

7, In the lane 314, the authors described as “modifying the membrane thickness did not significantly alter Gi2 activation”. Do the authors have any interpretation about the different activations between Gq and Gi2/o by the membrane thickness?

Answer: Our actual interpretation is that the membrane restricts the dynamics of the receptor, as recently proposed based on the cryoEM structure of the neurotensin receptor associated to Gi in lipid nanodiscs (Zhang et al. (2021). *Nat. Struct. Mol. Biol.* online ahead of print). Namely, in our case, the differences in membrane thickness appear to lead to differences in the amplitude of the TM6 movements, and this in turn has an impact of the way the G protein interacts with the receptor in a productive manner with regard to GDP-to-GTP exchange. Indeed, the major conformational state of active GHSR in thin membranes is characterized by an outward movement of TM6 of lower amplitude compared to that in thick membranes, based on LRET efficiency and previous MD simulations. We hypothesize that this small amplitude movement is sufficient to accommodate Gi, leading to its activation. This is consistent with the different cryoEM structures of GPCRs associated to distinct G protein subtypes where a small TM6 displacement is a hallmark of Gi coupling (Garcia-Nafria & Tate (2019). *Mol Cell Endocrinol* **488**, 1). To be noted, a GHSR conformational state with a larger displacement of TM6 such as that occurring in DOPC membranes still activates Gi, however, but with a slightly lower efficacy (see Fig.7d). This suggests that, besides selectivity, intrinsic efficacy at Gi activation could also be modulated, at least to some extent, by the amplitude of the TM6 movements. In contrast to Gi, the small TM6 displacement is not sufficient for Gq to interact with GHSR in an efficient manner in terms of GTP turnover, and it is only when the membrane thickness allows a TM6 movement of larger amplitude that GHSR activated Gq. In summary, in our model, the amplitude of the TM6 movements could be a component of the G protein-coupling selectivity and efficacy, and these movements could be in turn modulated by the intrinsic characteristics of the bilayer the receptor is immersed in. This has been added to the discussion section of the revised manuscript.

8, In figure 4b, it seems to me that there are statistical significance differences between the values of λ_{max} in the presence of PIP2 and no PIP2 in wild type and A204E by ghrelin.

Answer: there is indeed a statistical difference between the values of λ_{max} in the presence of PIP2 and no PIP2 for the wild type receptor, consistent with an impact of this lipid on the conformation of the wild type receptor. In contrast, the statistical analysis proposed by the reviewer revealed that the difference observed for the ligand-free A204E mutant was not significant, consistent with our model where PIP2 essentially impacts on the active state of GHSR. These statistics have been added to the revised version of **Figure 4**.

In addition, in figures 5a and 5b, are there any statistical significance differences between no PIP2 and the presence of PIP2 in (+GHSR+ghrelin)?

Answer: we repeated all these experiments twice again to improve the statistics. For the wild type receptor in the presence of ghrelin, the new data and statistical analyses shows that the difference between no PIP2 and PIP2 is significant. These statistics have been added to the revised version of the figure.

Also in the figure 6c and 6d, there are no statistical significance differences.

Answer: this was indeed not clear from our initial data. Hence, as for figure 5, we repeated all these experiments twice again to improve the statistics. The new data shows that there are statistically significant differences that favor our model where GHSR selectively activates Gi or Gq depending on the membrane thickness. These statistics have been added to revised version of **Figure 7**.

Methods for statistical analysis should be added.

Answer: as requested, the methods for statistical analyses have been added to legends of the corresponding figures in the revised version of our manuscript ("*Statistical values were obtained by means of unpaired Student's t test*").

9, Lane 527. PDB ID of NTSR1 should be added. The authors probably construct the GHSR model based on an inactive form of NTSR1.

Answer: in the initial version of the manuscript, the three-dimensional model of GHSR was indeed built from the crystal structure of the inactive form of NTSR1. However, as this was a concern for the other reviewers also, we performed all modeling again with the recently published structure of GHSR (PDB: 6KO5) instead of the NTSR1-based homology model. As a consequence, all the parts of the manuscript mentioning the use of NTSR1 as a template were removed and the pdb ID of the GHSR structure indicated.

Minor comments

1, Lane 333, Figure 6e? I think this is Figure 6d.

Answer: indeed, we confused the two panels of **Figure 6** (**Fig.7** in the revised version of the manuscript). This has been corrected in the revised version of the manuscript.

2, Lane 478. Please cut “in” from “described in”.

Answer: this has been changed in the revised version of the manuscript.

3, Lane 665, Figure legend of figure 4. “of” should be “or”.

Answer: this has been corrected in the revised version of the manuscript.

4, In the supplementary figures and legends, the alphabets should be changed to the small letters. In the text, the alphabets are described as the small letters. For example lane 80, there is “Fig. S1a” and lane 81 “Fig.S1b,c”.

Answer: This was indeed an oversight and we apologize for that. We have changed all the lettering of the figures in the supplementary section of the revised manuscript to be consistent with the main text.

5, In the figure S3, an oxygen is missing in the acyl chain of PI.

Answer: this has been corrected in the revised version of the manuscript.

6, Lane 91. MS should be described as MS (Mass Spectrometry).

Answer: this correction has been introduced in the revised version of the manuscript.

7, Lane 140, “(superscript numbers follow Ballesteros-Weinstein numbering)” should be move to the lane 93 after “cysteine mutant”.

Answer: this correction has been introduced in the revised version of the manuscript.

8, Lane 139. the BW number for R72 is missing.

Answer: the GPCRdb indicates R72 is located in ICL1, and thus provides no BW number for this residue. This is the reason we did not gave it a BW numbering.

9, In the text, the authors should unify the ghrelin receptor to GHSR.

Answer: this has been corrected in the revised version of the manuscript.

Reviewer #2

Major comments

1. Overall, these studies provide a persuasive account of the allosteric regulation of a GPCR by lipids and/or its membrane environment. My main concern is that the simulation studies are performed with a homology model of the GHSR rather than the recent structure. This may not be too much of a problem given the simulations are at CG resolution, but I think they should run additional simulations with the new GHSR structure to demonstrate this is indeed the case.

Answer: as this remark was also raised by the other reviewers, we repeated all our simulations with the crystal structure of GHSR (PDB: 6KO5) as a starting conformation instead of the NTS1R-based

model. Compared to what we described in the first version of the manuscript, we do not observe any significant difference in the distribution of the PIP2 around the receptor. More important, the binding sites we identified and the number of lipids they bind are the same, leading to conclusions similar to those initially drawn. As perfectly stated by the reviewer, this is perhaps not surprising, as we effectively work at the CG resolution. As new simulations have been performed, all parts of the manuscript mentioning the use of the NTSR1 receptor as a reference structure were removed and replaced by the new data and analyzes.

2. The receptor is immobilized on a solid matrix. Can we be certain that this will not perturb the conformational equilibria of the receptor, and the sensitivity of this to lipids?

Answer: our experimental procedure was not clearly described and we apologize for that. In fact, the receptor is immobilized on the matrix only during nanodisc assembly. It is then dissociated from the matrix and all subsequent fluorescence and functional experiments are carried out with the nanodisc particles in solution. The receptor is thus likely free to explore its conformational landscape under such conditions. In the same way, the lipids are also likely free to diffuse in the nanodisc.

3. Page 4: the FRET signal reaches a maximum at PIP2:GHSR of about 4 or 5. How does this compare with the mean number of PIP2 molecules bound per receptor in the simulations? (on page 5 it is stated this ratio is “fully consistent” with 5 or 6 PIP2 molecule bound in the simulations)

Answer: the maximal number of PIP2 molecules inferred from the FRET data to be bound to the receptor is about 5. If one takes into account the experimental error, this value is in the same range than that inferred from the simulations, where a mean of 6 PIP2 are predicted to be bound to one receptor at the same time. It cannot be excluded, however, that an additional PIP2 bound with lower affinity would appear in the CGMD simulations because of the particular characteristics of the method and would not be detected experimentally (see also answer to a point #6 below). Alternatively, this additional low affinity binding could contribute to the slight increase in the FRET signal we observed at higher PIP2-to-GHSR molar ratios (see also the answer to point 3 of reviewer #3, below). The similarity between computational and experimental outputs is also noticeable in the mutagenesis experiments where mutating the three sites identified in the computational analysis resulted in an almost total loss of the specific FRET signal. Moreover, mutating sites 1, 2 and 3 led to a ca. 50%, 40% and 20% decrease in the signal, respectively. This distribution is again consistent with the number of PIP2 molecules interacting with each site based on the CGMD analyses.

4. In the FRET experiments neither POPs or POPG competes with PIP2. Does either PS or PG binding to the receptor occur in the simulations?

Answer: this is an important remark, also with regard to the concern of reviewer #3 on the number of lipids we considered. As asked by the reviewer, we thus performed additional simulations with POPG. In these new simulations, 7 POPG molecules were added to each leaflet of the membrane in the presence or in the absence of the same number of PIP2 molecules. As shown in **Supplementary Figure 14** of the revised version of the manuscript, the distribution of the number of PIP2 adsorbed at the intracellular surface of GHSR is the same in the absence or in the presence of POPG. The same conclusion was drawn from the site-by-site analysis that showed no difference in PIP2 binding to sites 1, 2 and 3 when POPG was included in the membrane. This observation is

reinforced by the visual inspection of the resulting densities of POPG around GHSR, showing no significant binding of this lipid around the receptor at the same statistical threshold (**Supplementary Fig.13** in the revised manuscript). Taken together, these results confirm the FRET experiments and point to an absence of competition between POPG and PIP2 for binding the PIP2 sites we identified on GHSR, and, more generally, to an absence of specific interactions between the receptor and this lipid.

5. *The major conclusion from the FRET studies is that PIP2 binds preferentially to the active state of the GHSR (page 6). This is an important result. Interestingly it agrees well with the computational results in Ref 22 (for the A2AR) which predicted such a difference in PIP2 binding to active and inactive states, and so this should be mentioned.*

Answer: the reviewer is totally right in mentioning this similitude between our experimental data and previous pioneering computational data with A2AR that point to a similar mechanism. A sentence has been added to the discussion section of the revised version of the manuscript to take into account this similitude (“*Our data point to a mechanism where the effect of PIP2 occurs because of the preferential binding of this lipid to the receptor active conformation, so that this conformation would attract the lipid, in turn favoring the corresponding conformational state. Of importance, this is very similar to the mechanism initially proposed for A2AR based on computational studies where PIP2 also preferentially bound to the receptor active state¹⁷.*”.)

6. *Also, it should be possible to build a homology model of active state GHSR (especially as their simulations are based on a homology model of the inactive state) and show by simulations that PIP2 binds more strongly to the active state model, thus reinforcing this interpretation of their FRET studies.*

Answer: this is a very interesting remark and we thank the reviewer for raising it. Indeed, this generated additional interesting information we did not initially expect. As suggested by the reviewer, we performed all the simulations on two different conformations of GHSR. Each condition was tested 3 times, including those dedicated to delineate a possible competition between lipids. The “inactive” conformation directly corresponded to the X-ray structure deposited in the Protein Data Bank (PDB 6KO5) that described the receptor bound to a small antagonist. In addition, we generated an “active” conformation of GHSR by using a targeted MD protocol in which the final arrangements of the transmembrane domains to be reached was that of the dopamine receptor in the D2R:Gi complex (PDB: 6VMS). This procedure has been added to the methods section in the revised version of the manuscript. By doing so, an “active” (*i.e.*, coupled) conformation of GHSR was generated. The main difference between these two conformations is in the opening of TM5+TM6 segments and in the resulting elastic networks that allows the “active” conformation to open or close freely, whereas the “inactive” conformation was condemned to stay closed.

The graph in the new version of **Supplementary Figure 7** indicates no major difference between the distribution of PIP2 around the active and the inactive conformations of GHSR (black and grey histograms, respectively), with a mean of 6 PIP2 bound to the GHSR in both cases. PIP2 was thus still bound to the inactive state of GHSR in these simulations, in contrast to FRET where a significant decrease in the signal was observed for the A204E mutant and the LEAP2-loaded wild type receptor. A possible explanation for this discrepancy would be that the decreased affinity of PIP2 for its binding

sites in the receptor inactive state we invoke to explain the lower FRET signal under our experimental conditions would not be observed in the simulations because of the particular features of CGMD that make the lipids stick to the protein (Alessandri *et al.* (2019). Pitfalls of the Martini Model. *J. Chem. Theory Comput.* **15**, 5448), and thus do not allow any change in the protein:lipid exchange dynamics to be visualized. Interestingly, however, the site-by-site analysis nevertheless indicates a difference for site 3 (dark/light green histograms in **Supplementary Fig.7**). Indeed, this site is meanly occupied by one single PIP2 in both conformations but with a significant statistical decrease in the inactive conformation (from 60% to 40% of the whole conformations). This observation can be explained by a different position of PIP2 in the active conformation in which it intercalates between TM6 and TM7 (see **Supplementary Fig.11** in the revised manuscript). By doing so, PIP2 could impact on the relative movements of TM6, a hallmark of receptor activation. This feature is thus particularly interesting in the context of the experiments indicating that the binding of PIP2 could stabilize the active conformation of the receptor. These data and statements have been added to the revised version of our manuscript.

7. *The author tried to find a correlation between the membrane thickness and receptor activation efficiency by using only two lipid species of different lengths. Is it possible to explore an additional lipid with a tail length other than 14C to 18C to make this correlation more persuasive?*

Answer: to address this remark, we analyzed G protein activation with the receptor embedded in nanodiscs composed of lipid with an intermediate chain length, *i.e.*, 16:1 ($\Delta 9$ -Cis) PC. As shown in the revised manuscript, the profile obtained is closely related to that observed in 14:1 ($\Delta 9$ -Cis) PC, although a slight Gq activation could be observed in this case. This is in line with our previous observations indicating that Gq selectivity is related to membrane thickness, with a change occurring for a narrow window of lipid tail length. These data have been added to the supplementary section of the revised version of the manuscript (**Supplementary Fig.17**).

8. *(Line 279 P.8) The authors suggest that DMoPC (14C) and DOPC (18C) should have the same membrane fluidity because they display the same unsaturation degree. However, lipid tail length plays a major part in determining the membrane fluidity as well.*

Answer: membrane fluidity is indeed dependent also on the length on the lipid tail, and we apologize for our understatement. To check whether the fluidity was the same in both cases, we added to the revised version of the manuscript a new series of experiments where laurdan was introduced into DMoPC and DOPC nanodiscs. Laurdan is a fluorophore that inserts in the bilayer and is commonly used to monitor membrane fluidity, as a general polarization (GP) value can be measured from the emission intensity of this compound that directly reflects the fluidity of the membrane. A very similar GP value was obtained for both DMoPC and DOPC nanodiscs (-0.017 and -0.018, respectively), indicating that the bilayer fluidity is similar in both cases. These values are given in **Table S3** of the revised version of the manuscript.

Minor comments

Some work is needed to make Fig. 2 a bit clearer, especially to the non-specialist reader. Specifically:

- Fig 2a/b – It is unclear what this density is supposed to show. It looks like the authors need to change the isovalue of the density in the VMD settings. This also looks like the ‘density’ has been

calculated on a single simulation snapshot rather than by averaging over the trajectory but this might just be a problem with the selected isovalue.

Answer: We completely remastered this figure to show the binding of PIP2 around GHSR at different isovalues, using a more classical orientation of the receptor that depicts more clearly the specificity of binding of PIP2 on the intracellular side of the receptor. The densities represented were computed using the volmap tool of VMD selecting all the frames of one of the three simulations performed on the active conformation of GHSR. We also added new inserts to this figure showing the reproducibility of these densities among the three simulations for both the inactive and active conformations of GHSR. For the sake of clarity, these representations have been removed from the main text where **Figure 2** now only shows the position of PIP2 in its three binding sites; they are instead given in **Supplementary Figure 6** of the revised version of the manuscript.

• Fig 2c-e – This might be clearer if the authors show the PIP2 using the cg_bonds script in MARTINI rather than as unconnected VDW spheres. Also, specify the colours used in the figure legend.

Answer: as requested, the PIP2 representation was changed using cg_bonds in the new version of **Figure 2**. The colors used were also indicated in all legends.

Reviewer #3

1/ The first issue is the scope/structure of the paper. The title "Allosteric modulation of ghrelin receptor 1 signaling by lipids" is stricto sensu correct (PIP2 is a lipid indeed) but seems to suggest that the authors will investigate the effect of various lipids while they focus on PIP2. Considering that, along with PIP2, other lipids can modulate GPCRs, it is not quite clear while they aren't included. The specific displacement of the labeled PIP2 by unlabelled PIP2 but not by other unlabelled lipids indicate that PIP2 indeed interacts but it does not mean that other lipids do or do not.

Answer: our manuscript considered the effect of PIP2 and membrane thickness only. This indeed appears as a limited set of lipids considering the diversity of the plasma membrane. Because of the large number of experiments that are required to reach a convincing conclusion for the impact of a given lipid, an extensive analysis on many different compounds appeared complicated within the timeframe of the revision of the present manuscript. We have nevertheless some indications for other compounds. First, as stated by the reviewer, no significant interaction was observed for other lipids such as POPG, at least based on the competition experiments with PIP2 (and now GM3). This was confirmed by our new computational data indicating that this lipid displays no specific binding to GHSR (see **Supplementary Figs.13,14**). Then, to further address the reviewer remark, we added to the revised version of our manuscript a first series of new experiments with another compound, the ganglioside GM3. We used this lipid as it had been proposed to selectively interact with different membrane receptors and in particular to GPCRs, based on prominent computational studies (Ansell et al. (2020). *Biophys. J.* **119**, 300; Song et al. (2019). *Structure* **27**, 392). Moreover, it had also been shown to be involved in biological processes that are under the control of the ghrelinergic system (Dieterle et al. (2020). *Int. J. Obes.* **44**, 510) and could thus be physiologically relevant to the present situation. This new data is detailed in the answer to point #11 below.

In terms of structure, I find somewhat unclear why, after focusing on PIP2, the authors then look at changing the bilayer thickness. These seems like somewhat separated studies glued together.

Answer: we totally agree with the reviewer that the two parts of the study are unrelated. In fact, what we wanted to do in our manuscript was to address how the different properties of the bilayer could possibly impact on GHSR functioning. It had been shown that lipids affect membrane proteins into essentially two different ways, *i.e.*, through specific interactions and through bulk physicochemical properties (thickness, lateral pressure, curvature...). We thus intended to provide an example of both effects in this paper. For the specific interaction, we focused on PIP2 (and GM3 in the revised version), as (i) these lipids have been proposed to selectively interact with GPCRs and (ii) the role of other particular lipids, in particular cholesterol, had already been more extensively analyzed, including by our group (*e.g.*, Casiraghi et al. (2016) *J. Am. Chem. Soc.* **138**, 11170). For the membrane bulk properties, as the effect of features such as curvature or lateral pressure could not be addressed with the nanodisc model system, we focused on thickness. These are the reasons we thought it would be of interest for the two topics, although not related, to be brought together in the same manuscript, as they provide a more general picture of the different ways the membrane could regulate ghrelin signaling. We hope the reviewer will find it relevant.

2/ Can the authors provide some information regarding the labelled analogs. I have been struggling in looking for the exact structure of the labeled lipids on the thermofisher website, but I assume they in fact come from Echelon where BODIPY-FL PIP2 contains two peptide bonds. From what I could find the BODIPY-FL PA does not. This could affect the integration in the bilayer, the location of the probe and therefore the FRET signal. Could that affect the quantum yield of the probes (I guess that Figure S2 suggests it doesn't)?

Answer: the details on the labeled compounds were indeed missing. Also, the provider of BODIPY-FL PIP2 was mistaken. We deeply apologize for that. BODIPY-FL PIP2 was indeed from Echelon. Only BODIPY-FL PA was from Thermofisher (catalog number D3805), but is no more commercialized by this supplier. As perfectly noted by the reviewer, BODIPY-FL PA does not contain a peptide bond, in contrast to BODIPY-FL PIP2, and, in addition, differs in the position of the fluorophore (see molecular structures below; these structures are now provided in the **Supplementary Fig.4**). This could indeed affect the way the labeled lipid inserts into the bilayer and thus the FRET signal. To rule out this possible mechanism, we carried out an additional control with BODIPY-FL PI (Echelon). This compound was selected as (i) PI did not compete with PIP2 in our initial FRET assays (**Fig.1b** in our manuscript) and (ii) it includes the same labeled acyl tail than BODIPY-FL PIP2 (see below). Both compounds should therefore insert into the bilayer in a similar manner. As shown in **Supplementary Figure 3** of the revised manuscript, essentially no specific FRET signal was observed with BODIPY-FL PI, indicating that the difference between labeled PIP2 and the control lipids are not likely due to a difference in the way the probe inserts into the lipid nanodiscs. As far as the quantum yield is considered, as perfectly noted by the reviewer, insertion into the bilayer does not significantly affected the quantum yield of the probe compared to the same probe solubilized in a detergent solution, based on the similar fluorescence properties in both situations (**Supplementary Fig.2**).

Structure of the commercial BODIPY-FL analogs of PA (a), PIP2 (b) and PI (c) used in this work. These structures are from the supplier's catalogs and are provided in **Supplementary Figure 4** of the revised manuscript.

3/ On Figure 1a, for lipid:protein ratio above 5 it appears that this fluorescence value goes up with "a slope similar to that of the control lipid Bodipy-FL PA, suggesting a random distribution of the additional fluorescent PIP2 molecules in the nanodisc". This seems odd to me. Shouldn't we assume that if all three binding sites are occupied maximum FRET efficacy is achieved or that, at least, distantly located FRET acceptors should not contribute ?

Answer: Indeed, this is a puzzling observation. A maximum FRET ratio value should have been observed upon full occupation of the PIP2 binding sites. This was not the case experimentally, as a slight increase in the FRET signal was observed at higher PIP2-to-lipid ratios, although at these higher ratios no additional structural or functional effects were observed. We have no firm explanation for this observation. A possibility would be to consider very low affinity sites with a very transient interaction that would give rise to pseudo non-specific binding. These transient binding events are observed in some of the CGMD simulations, indicating the possible occurrence of such low affinity binding sites (see also answer to point 3 of reviewer #2). Another possibility that cannot be excluded at this stage would be to consider some sort of crowding effects because of the confined nature of the nanodiscs, where long range, low efficiency FRET effects would occur upon increasing the concentration of labeled lipids in the disc. Both effects could account also for the low FRET signal observed with BODIPY-FL PA or PI. In any case, and we hope the reviewer will agree with us, this does not affect the main conclusion inferred from the FRET-based assay on the occurrence of specific binding sites for PIP2 on GHSR. A sentence has been added to the revised version of the manuscript to take into account this remark (*"Above this value, the signal increased with a slope similar to that of the control lipid Bodipy-FL PA, suggesting a random distribution of the additional fluorescent PIP2 molecules in the nanodisc. The reason for this slight increase is not clear at the present stage of the analysis, but it could be due to very transient interactions between BODIPY-FL PIP2 and low affinity sites, as observed in the computational analyses (see below), or non-specific effects associated to a crowding of the labeled lipids near the receptor because of the confined nature of the nanodisc structure."*)

4/ The choice of Lumi4-Tb as FRET donor seems odd to me. Sure, the extended lifetime allows to decrease background, but the probe is very large and, being directly coupled to the bottom of TM6 one could fear that it may affect the dynamics of the receptor. The authors should at least comment on that.

Answer: The choice of Lumi4-Tb as FRET donor is indeed not the most common one. We nevertheless found that using luminescent lanthanides as a donor in LRET measurements had some technical advantages over conventional FRET with smaller fluorophores when used as a complementary readout to MB fluorescence to provide indications on the geometrical arrangement of the receptor and receptor:G protein complexes. One of the main characteristics of LRET is the possibility to determine luminescence lifetime of the donor involved in energy transfer, a parameter essential for distance calculation, from the analysis of sensitized acceptor emission decay. It is thus possible, using this sensitized acceptor emission, to measure efficiency of energy transfer even when the sample is incompletely labeled (see Selvin (2002). *Annu. Rev. Biophys. Biomol. Struct.* **31**, 275 for a review). This appears especially relevant to membrane proteins embedded in lipid bilayers, as the latter can significantly affect labeling, as properly noted by the reviewer in a subsequent remark. Accordingly, LRET has been often used with different membrane proteins and has been shown to be an appropriate tool for describing their dynamics. We ourselves extensively developed and used this approach with the ghrelin receptor to analyze its conformational dynamics and interactions with G proteins (Damian et al. (2015) *Proc Natl Acad Sci USA* **112**, 1601; Damian et al. (2018) *Proc Natl Acad Sci USA* **115**, 4501). We thus built here upon this previous work, as it had allowed us to develop and rationalize all the labeling, analysis and interpretation procedures. Nevertheless, as perfectly noted by the reviewer, it cannot be excluded that the large size of the probe affects, to some extent, the intrinsic dynamics of GHSR when bound to the tip of TM6. To be noted, the same probe had been previously introduced in a closely related position of another GPCR, the vasopressin V2R receptor (position 6.31), and had been shown to appropriately report on its dynamics also (Rahmeh et al. (2012) *Proc. Natl. Acad. Sci. USA* **109**, 6733). Besides, what we monitored here were the relative differences in the LRET signal for the same protein construct, the Lumi Tb-labeled receptor, in different lipid environments, so that a possible impact of the probe on the absolute values of the dynamics should not alter our conclusions. A sentence noting this caveat has been added to the revised version of the manuscript (*"To be noted, although it cannot be excluded that the size of the Lumi-4 probe may affect the absolute values of the receptor dynamics, we nevertheless previously showed that LRET was well-adapted to monitor the changes in GHSR conformation in a variety of conditions"*).

Regarding LRET, the authors also rely on the reader's preliminary knowledge of their previous (2015?) papers to know what LRET means/is and permit to reveal. A bit of more explanation on the information provided by the sensitized-emission decay would be useful as it is far from intuitive.

Answer: as requested by the reviewer, we added explanations on the information provided by the sensitized-emission decay in the results section of the revised version of our manuscript.

5/ The authors conclude that PIP2 favors the active state and show that, conversely, "PIP2 preferentially binds to the active state of GHSR" by showing lack of interaction on a mutant devoid of constitutive activity. Wouldn't this better demonstrated by monitoring GSHR:PIP2 FRET signal in

the absence and presence of the inverse agonist LEAP2? This appears to be missing from Figure 4.

Answer: this is a very good suggestion, and we fully agree this was missing from our manuscript. We carried out the proposed experiment. The FRET signal originating from PIP2:GHSR interactions was significantly decreased when LEAP2 was bound to the wild type receptor, consistent with a model where PIP2 would preferentially bind to the active state of GHSR. This data has been added to **Figure 4** in the revised version of the manuscript.

6/ The effect of PIP2 on G coupling is interesting but somewhat weak (especially in light of the MB data). It seems that the differences in GTP turnover are not statistically significant. If so, the authors should comment.

Answer: Indeed, this was not clear from our previous set of data. As stated in the answers to reviewer #1, we carried out additional series of GTP turnover experiments to get better statistics. As shown in **Figure 5**, these data, which are now mean \pm SD of five experiments, show that the difference in GTP binding \pm PIP2 is statistically significant. These statistics have been added to the revised version of the manuscript.

7/ Regarding the cytoplasmic vs extracellular location of the binding sites, Fig S4 lacks important controls/information. The methods do not specify how lumi4-Tb was attached to either Cys255 or Cys304 but I assume by a thiol-reactive group such as maleimide. The labelling details are not provided either.

Answer: the labeling details were missing in the previous version of our manuscript and we apologize for that. As perfectly assumed by the reviewer, attachment of the fluorophore was carried out using a maleimide derivative of the probe with a receptor mutant containing a unique reactive cysteine at position 304^{7,34}. This mutant was previously devised for labeling GHSR with monobromobimane and was shown to allow a specific labeling at this position while preserving the pharmacological properties of the receptor (Damian et al. (2012). *Proc. Natl. Acad. Sci. USA* **109**, 8304). These details have been added to the Methods section of the revised version of the manuscript.

In my experience that reactivity of target cysteine to labels vary greatly, sometimes as much as 1 in 10. I also note that Cys304 appears to be located in/at the edge of the bilayer and thus poorly accessible. The authors much show labelling and fluorescence of C304 and not just lack of transfer.

Answer: Indeed, the reactivity of cysteine residues towards maleimide probes is very dependent on their proximity to the membrane, and it can significantly decrease when these residues are close to the bilayer. Hence, an inefficient labeling could have been a plausible explanation for the absence of signal when the probe was attached to extracellular part of the receptor, as C304^{7,34} is indeed close to the interface with the membrane. As requested by the reviewer, we present in the revised version of the manuscript the absorption and emission spectra of the modified protein labeled at position 304^{7,34} with Lumi-4 Tb (**Supplementary Fig.5a,b**). Both spectra are indicative of an efficient labeling of the protein. Besides, in the new set of data we added to the revised version of the manuscript, an efficient transfer signal was observed between GHSR labeled at this same position and fluorescent GM3 (**Fig.6a**). This observation further confirms that C304^{7,34} is labeled with the

Lumi-4 Tb probe, and that the reduced FRET signal with PIP2 is therefore not likely to result from the absence of the fluorescence donor on the receptor.

8/ Model vs. crystal. I am surprised that the authors did not correct their modelling by using the recently published structure of GSHR (though I understand that it requires extra work). There are some differences between NTS1 and GSHR at the proposed PIP2 binding sites that would justify re-running the coarse-grained simulations. On the other hand, the authors may argue that the model was “good enough” to identify experimentally-validated PIP2 binding sites, but then again they could have tested the same sites with a literature survey (which they did perform).

Answer: to address this remark and the similar ones from the other reviewers, all the simulations were performed again with the crystal structure of the antagonist-bound GHSR instead of the NTS1R-based model. This structure was used as an “inactive” conformation. Additionally, as suggested by reviewer #2, we also performed simulations on an “active” conformation artificially built by comparative modelling with the D2R:Gi complex as a reference structure. The major difference between these two conformations was the possibility for the latter to open and close freely, whereas the inactive conformation stayed closed because of the CG model used. The literature could have indeed helped to identify some of the binding sites for PIP2, but our computational work helped us bring additional important information. As perfectly noted by the reviewer, as far as the binding sites are concerned, there are subtle differences in the residues involved compared to other receptors such as NST1R (see also the answer to the first remark of reviewer #1), and identifying these new sites would have been hardly possible without the CGMD simulations.

9/ *The physiological relevance of the study on the effect of bilayer thickness remains unclear to me. To my knowledge it is not expected that bilayer thickness will change significantly from one cell type to another, whether through acyl chain length distribution or temperature changes.*

Answer: it is not indeed expected that the global thickness of the lipid bilayer changes from one cell type to another. The great diversity of lipid species nevertheless suggests that a wide range of combinatorial interactions occurs between them in cell membranes, so that the latter are not likely homogeneous with regard to their physicochemical properties. Current data support the concept that relatively small changes in lipid composition have nevertheless a dramatic effect on bilayer properties. In particular, many local variations in composition and in asymmetric organization of the leaflets lead to microdomains with differences in their physicochemical features, including their thickness. For instance, lipid microdomains rich in cholesterol and sphingolipids have been shown to be associated with an increase in the ordering of adjacent lipid acyl chains and thus in the membrane hydrophobic thickness. This is not only a matter of the plasma membrane, as the latter and the intracellular compartments differ in their lipid composition and physicochemical features, and discrete domains with increased bilayer thickness have been recently identified in the ER membrane that contribute to modulation of the lateral diffusion of membrane proteins (Prasad et al. (2020) *Sci. Adv.* **6**, eaba 5130). This may have an importance with regard to the increasing evidence for selective GPCR signaling in intracellular compartments. Finally, cells are submitted to diverse stresses, in particular mechanical stresses, and several GPCRs have been proposed to function as mechanosensors (see for instance Erdogmus et al. (2019). *Nat. Commun.* **10**, 5784 and references therein). Interestingly, at the molecular level, modulation in plasma membrane tension modulates its physicochemical features, and in particular its local thickness (Le Roux et al. (2019) *Philos Trans R*

Soc Lond B Biol Sci. **374**:20180221). Hence, although this may not be a major process, local changes in the thickness of a given cell membrane could nevertheless occur that may affect to some extent receptor lateral diffusion, localization, conformational dynamics and signaling. To take into account this *caveat*, the ending paragraph discussing this topic has been shortened.

10/ The Discussion should be shortened. The first part is just a repeat of the Result section. The last paragraphs are generalities on membrane variations. I don't think that Figure 7 brings much.

Answer: as requested by the reviewer in this remark and in the previous one, the discussion has been shortened and **Figure 7** removed from the revised version of the manuscript.

11. Overall, I think this is a nice study but the focus should be clearer. I personally would have liked to see other lipids being tested as I'm fearing that importance PIP2 is being over represented in the field.

Answer: we thank the reviewer for his nice appreciation of our work. We agree with him that a lot of emphasis has been put on PIP2 lately and that these analyses are to be extended to a variety of other lipids as well, if one considers the diversity in the composition of plasma membranes. As stated above, other lipids such as PI, PIP3 and POPG did not seem to specifically interact with GHSR, based on our FRET experiments and on the new computational data that has been added to the revised version of the manuscript. This does not exclude, however, that these lipids affect the general properties of the bilayer such as its charge and, by doing so, also affect receptor functioning, as shown for other GPCRs (Strohman et al. (2019). *Nature Comm.* **20**, 2234; Inagaki et al. (2012). *J. Mol. Biol.* **417**, 95). As state above, because of the large number of experiments that are required to reach a convincing conclusion for the impact of a given lipid on receptor functioning, an extensive analysis on many different lipids appeared complicated within the timeframe of the revision of the present manuscript. As a first step along these lines, in addition to the new simulations with POPG, we nevertheless tried to address the effect of another particular lipid, GM3. The latter was selected as it had been very convincingly shown in computational studies to interact with other GPCRs (Ansell et al. (2020). *Biophys. J.* **119**, 300; Song et al. (2019). *Structure* **27**, 392). In addition, this class of lipids has emerged as a novel important regulator of metabolism, glucose homeostasis and body weight, all functions that are highly relevant to GHSR signaling (Dieterle et al. (2020). *Int. J. Obes.* **44**, 510). By doing so, we now show that GM3 also selectively interacts with GHSR, in particular in its extracellular region, a region where PIP2 does not bind. This distribution of lipid binding sites is consistent with the asymmetry of the plasma membrane, as GM3 is only found in the extracellular leaflet whereas PIP2 is exclusively located in the intracellular one. By binding the receptor, GM3 also seemed to affect GHSR-mediated G protein activation, although through a mechanism that may be subtly different from that involved with PIP2. Indeed, GM3 binding to the extracellular regions of GHSR seemed to mostly affect agonist-mediated activation whereas PIP2 binding to the intracellular part of the receptor impacted both on basal- and agonist-dependent processes. This first series of data with GM3 has been added to the revised version of the manuscript. Taken together, although we fully agree it still does not indeed bring an extensive panorama of the effect of all the different lipids present in the plasma and intracellular membranes, we hope our data nevertheless represents a first step towards a better understanding of the complex allosteric modulation of GHSR functioning by its lipid environment.

Reviewer #1 (Remarks to the Author):

I have checked the revised manuscript by Damian et al. In the manuscript, the authors have truly addressed all comments made during the initial submission. In particular, I appreciate the new data with GM3 and the application of the recently published GHSR structure instead of NTSR1. I think this manuscript will have a possibility to be accepted by Nature Communications.

Reviewer #2 (Remarks to the Author):

Review of NCOMMS-20-42334A

This is a revised version of a manuscript on allosteric regulation of the ghelin receptor by lipids, which is an important and timely topic. There is accumulating evidence for regulation of various GPCRs by lipids, but the mechanism and functional role of such regulation remains unclear.

The study combines experimental (using FRET and nanodiscs) and computational (MD simulation based) approaches.

The authors have addressed the major points which I previously raised.

1. The simulation studies have been repeated with the crystal structure of GHSR (PDB 6KO5) rather than a homology model. This provides a much greater degree of certainty in this aspect of the results.
2. The role of the solid matrix in nanodisc assembly has been clarified.
3. The comparison between experiment and simulation with respect to the numbers of bound PIP2 molecules is now clearer.
4. It has been demonstrated that PG does not appear to compete with PIP2 in the simulations, which parallels the experimental observations.
5. It is now noted that for both GHSR and the A2AR, experiments and/or simulations suggest preferential binding of PIP2 to the active state of the receptor. However, for the GHSR a difference in the number of PIP2s bound in was not seen when comparing simulations of the inactive state (6KO5) and of an active state model. I am not sure the lack of a difference reflects the limitations of the Martini 2 model, but once PIP2 is included in Martini 3 (this is not yet the case) it may in the future be possible to test this.
6. The experiments on the effect of bilayer thickness have been refined, adding to the strength of the argument.
7. Laurdan fluorescence measurements of membrane fluidity have been added.
8. A number of the figures have been made clearer.

Overall, I am satisfied with the authors' responses and changes, and I am happy to recommend that the ms. should be accepted for publication.

Reviewer #3 (Remarks to the Author):

The revised manuscript by Damian et al. is improved and include a number of new experiments, as requested.

Overall, my comments and questions have been answered.

The main point remains the overall consistency of the manuscript, with a particular choice of (focusing on PIP2 and now GM3) together with the inclusion of the thickness study. The choice of GM3 is somewhat strange, especially since cholesterol is mostly left out.

That being said, the experiments are well crafted and convincing, the data, and their interpretation are sound.

In the end, it is now purely an editorial choice to decide whether the title should be adapted, or things left out, but from a scientific standpoint the study is convincing and solid and I have no more critique to address.

REVIEWERS' COMMENTS

Reviewer #1 (Remarks to the Author):

I have checked the revised manuscript by Damian et al. In the manuscript, the authors have truly addressed all comments made during the initial submission. In particular, I appreciate the new data with GM3 and the application of the recently published GHSR structure instead of NTSR1. I think this manuscript will have a possibility to be accepted by Nature Communications.

Reviewer #2 (Remarks to the Author):

Review of NCOMMS-20-42334A

This is a revised version of a manuscript on allosteric regulation of the ghrelin receptor by lipids, which is an important and timely topic. There is accumulating evidence for regulation of various GPCRs by lipids, but the mechanism and functional role of such regulation remains unclear.

The study combines experimental (using FRET and nanodiscs) and computational (MD simulation based) approaches.

The authors have addressed the major points which I previously raised.

1. The simulation studies have been repeated with the crystal structure of GHSR (PDB 6KO5) rather than a homology model. This provides a much greater degree of certainty in this aspect of the results.

2. The role of the solid matrix in nanodisc assembly has been clarified.

3. The comparison between experiment and simulation with respect to the numbers of bounds PIP2 molecules is now clearer

4. It has been demonstrated that PG does not appear to compete with PIP2 in the simulations, which parallels the experimental observations.

5. It is now noted that for both GHSR and the A2AR, experiments and/or simulations suggest preferential binding of PIP2 to the active state of the receptor. However, for the GHSR a difference in the number of PIP2s bound in was not seen when comparing simulations of the inactive state (6KO5) and of an active state model. I am not sure the lack of a difference reflects the limitations of the Martini 2 model, but once PIP2 is included in Martini 3 (this is not yet the case) it may in the future be possible to test this.

6. The experiments on the effect of bilayer thickness have been refined, adding to the strength of the argument.

7. Laurdan fluorescence measurements of membrane fluidity have been added.

8. A number of the figures have been made clearer.

Overall, I am satisfied with the authors' responses and changes, and I am happy to recommend that the ms should be accepted for publication.

Reviewer #3 (Remarks to the Author):

The revised manuscript by Damian et al. is improved and include a number of new experiments, as requested.

Overall, my comments and questions have been answered. The main point remains the overall consistency of the manuscript, with a particular choice of (focusing on PIP2 and now GM3) together with the inclusion of the thickness study. The choice of GM3 is somewhat strange, especially since cholesterol if mostly left out. That being said, the experiments are well crafted and convincing, the data, and their interpretation are sound. In the end, it is now purely an editorial choice to decide whether the title should be adapted, or things left out, but from a scientific standpoint the study is convincing and solid and I have no more critique to address.

ANSWERS TO REVIEWERS' COMMENTS

Reviewer #3 (Remarks to the Author):

Overall, my comments and questions have been answered. The main point remains the overall consistency of the manuscript, with a particular choice of (focusing on PIP2 and now GM3) together with the inclusion of the thickness study. The choice of GM3 is somewhat strange, especially since cholesterol if mostly left out. That being said, the experiments are well crafted and convincing, the data, and their interpretation are sound. In the end, it is now purely an editorial choice to decide whether the title should be adapted, or things left out, but from a scientific standpoint the study is convincing and solid and I have no more critique to address.

Answer: we addressed the case of GM3 instead of cholesterol, as the effect of the latter on GPCR functioning had been much more extensively addressed in the literature recently. To be noted, we nevertheless report in the present manuscript some preliminary evidence showing that cholesterol likely impacts of GHSR conformation also, as it does for these many other receptors (Supplementary figure 10).

For the title, we considered that our manuscript addressed the effects of lipids through several examples, either of specific interactions (PIP2, GM3) and of bulk effects (membrane thickness). This is the reason why we thought we could use a general title that states the allosteric effects of lipids instead of that of PIP2, GM3 and membrane thickness strictly, although we fully agree with the reviewer that our manuscripts does not report *stricto sensu* on the effects of lipids from a general point of view. However, as the reviewer states, if the title should be adapted for editorial reasons, then "Allosteric modulation of ghrelin receptor signaling by lipids" could be replaced by "Allosteric modulation of ghrelin receptor signaling by PIP2, GM3 and membrane thickness".